# Bidirectional encoding of motion contrast in the mouse superior colliculus

**Jad Barchini[1,2], Xuefeng Shi[1,3,4], Hui Chen[1,5,6], Jianhua Cang[1,5,6]***

[1]Department of Neurobiology, Northwestern University, Evanston, United States; [2]Interdepartmental Neuroscience Program, Northwestern University, Evanston, United States; [3]Tianjin Key Laboratory of Ophthalmology and Visual Science, Tianjin Eye Institute, Clinical College of Ophthalmology, Tianjin Medical University, Tianjin, China; [4]Department of Pediatric Ophthalmology and Strabismus, Tianjin Eye Hospital, Tianjin, China; [5]Department of Psychology, University of Virginia, Charlottesville, United States; [6]Department of Biology, University of Virginia, Charlottesville, United States

**Abstract** Detection of salient objects in the visual scene is a vital aspect of an animal's interactions with its environment. Here, we show that neurons in the mouse superior colliculus (SC) encode visual saliency by detecting motion contrast between stimulus center and surround. Excitatory neurons in the most superficial lamina of the SC are contextually modulated, monotonically increasing their response from suppression by the same-direction surround to maximal potentiation by an oppositely-moving surround. The degree of this potentiation declines with depth in the SC. Inhibitory neurons are suppressed by any surround at all depths. These response modulations in both neuronal populations are much more prominent to direction contrast than to phase, temporal frequency, or static orientation contrast, suggesting feature-specific saliency encoding in the mouse SC. Together, our findings provide evidence supporting locally generated feature representations in the SC, and lay the foundations towards a mechanistic and evolutionary understanding of their emergence.
DOI: https://doi.org/10.7554/eLife.35261.001

*For correspondence:
cang@virginia.edu

**Competing interests:** The authors declare that no competing interests exist.

## Introduction

The detection of objects in the environment is crucial for an animal's ability to efficiently and safely navigate the world. In the visual system, objects are processed by neurons that respond to specific features in their receptive fields (RFs), such as orientation, movement direction, luminance, and color. Being spatially restricted, each RF provides a local representation of the visual scene. At the perceptual level, however, the same stimulus presented within an RF could appear drastically different depending on its context. For example, a vertical bar would 'pop out' perceptually when it is surrounded by horizontal bars, but not among other identical vertical bars (*Li, 1999*). Such saliency computation thus requires a comparison between local and global visual features at the neuronal level.

Most studies on feature-specific saliency computation have been conducted in primate and cat primary visual cortex (V1), predominantly in the context of orientation selectivity. It was shown that V1 neurons are suppressed by stimuli of the same orientation in regions surrounding the RF (*Jones et al., 2002*; *Knierim and van Essen, 1992*; *Sengpiel et al., 1997*), consistent with the classical surround suppression seen at the level of the retina and lateral geniculate nucleus (*Sachdev et al., 2012*). Importantly, V1 neurons displayed lower levels of suppression when static gratings of orthogonal orientations were shown in the surround (*Kastner et al., 1999*; *Knierim and van Essen, 1992*; *Nothdurft et al., 1999*). In response to moving gratings, orthogonal surrounds

were even able to induce a certain level of response facilitation in primate V1 (*Jones et al., 2001*; *Jones et al., 2002*; *Sillito et al., 1995*). In other words, depending on the relationship between the properties of the center and surround stimuli, differential levels of suppression or facilitation can occur, thus providing a neural basis for the perceptual 'pop-out' phenomenon.

It is theorized that feature-specific saliency computations are combined into a map to represent the total saliency value of each point in space (*Veale et al., 2017*). Although the exact location of where the saliency map is first generated is still a matter of debate, there is a general agreement that the superior colliculus (SC) in the midbrain contains such a map (*Veale et al., 2017*; *Zhaoping, 2016*). In primates, SC neurons are not tuned to specific visual properties, consistent with the notion of feature-agnostic saliency representation (*White et al., 2017*). In contrast, in lower vertebrates such as fish and birds, where neocortex has not evolved, neurons in the optic tectum, the homologue of the mammalian SC, can perform certain feature-specific saliency computations (*Ben-Tov et al., 2015*; *Frost et al., 1981*; *Sun et al., 2002*; *Zahar et al., 2012*). This has led to the idea that the locus of saliency computation has migrated evolutionarily, among many other visual system functions, from the tectum to the visual cortex (*Zhaoping, 2016*).

These considerations thus raise an intriguing question about saliency computation in mice. Although neurons in mouse V1 show similar selectivity compared to those in higher mammals (*Niell and Stryker, 2008*), the SC remains the most prominent retinal target in mice, and mediates visually-guided behaviors (*Ellis et al., 2016*; *Liang et al., 2015*; *Shang et al., 2015*; *Wei et al., 2015*; *Zhao et al., 2014*). Unlike in primates, most visual neurons in the mouse SC are tuned to features such as motion direction or stimulus orientation (*Gale and Murphy, 2014*; *Wang et al., 2010*). Here, we study how neurons in the *stratum griseum superficiale* (SGS) of the mouse SC respond to several types of motion contrast between RF center and surround. We use 2-photon calcium imaging to first study direction contrast in a highly selective population of SGS neurons in the superficial SGS (sSGS). Importantly, by imaging in transgenic mice with labeled GABAergic neurons, we reveal striking differences in the responses of excitatory and inhibitory neurons to direction contrast. We then demonstrate a bias in sSGS neuronal populations towards encoding direction contrast, as opposed to other features of the moving stimulus, possibly identifying a feature-specific saliency encoding in these superficial neurons. Finally, we delve deeper into the SGS to describe the depth-dependent profile of direction contrast responses. Together, our findings provide important information on how motion contrast is encoded by visual neurons in the SC.

## Results

### Responses of sSGS excitatory neurons are modulated by motion contrast

We first performed 2-photon calcium imaging of SGS neurons in anesthetized mice using the synthetic calcium dye Cal-520. Visual cortex was removed to allow optical access and eliminate any potential cortical influence on SC responses (*Figure 1A*). We specifically imaged the most superficial lamina of the SGS (sSGS,<50 μm from the surface), which we have recently shown to be enriched with neurons that are highly selective for movement direction (*Inayat et al., 2015*). Here, we uncoupled the movement direction between the stimulus center and surround, and determined how sSGS responses were modulated by this form of motion contrast (*Figure 1C*). These experiments were performed in mice where GAD2$^+$ neurons were fluorescently labeled with tdTomato, allowing us to compare the response properties of inhibitory (GAD2$^+$, GABAergic) and excitatory (GAD2$^-$) neurons (*Figure 1B*).

For each imaging session, we first mapped the RFs of the imaged neurons using a flashing black square on a gray background (*Figure 1—figure supplement 1*). A small circular patch of gratings (10° radius) was then placed at the center of the RFs and drifted in different directions, to determine the tuning and preferred direction of each imaged neuron. Consistent with our previous finding, about half of sSGS neurons were responsive to the drifting gratings (n = 355/811 GAD2$^-$ and n = 379/652 GAD2$^+$, from 9 mice, see Materials and methods for details of determining responsiveness), and most of them were direction selective (DS), showing much higher increases in fluorescence to their preferred direction than to the opposite direction (*Figure 1C*; 259/355, 73.0% of GAD2$^-$, and 239/379, 63.1% of GAD2$^+$ neurons had gDSI $\geq$0.25).

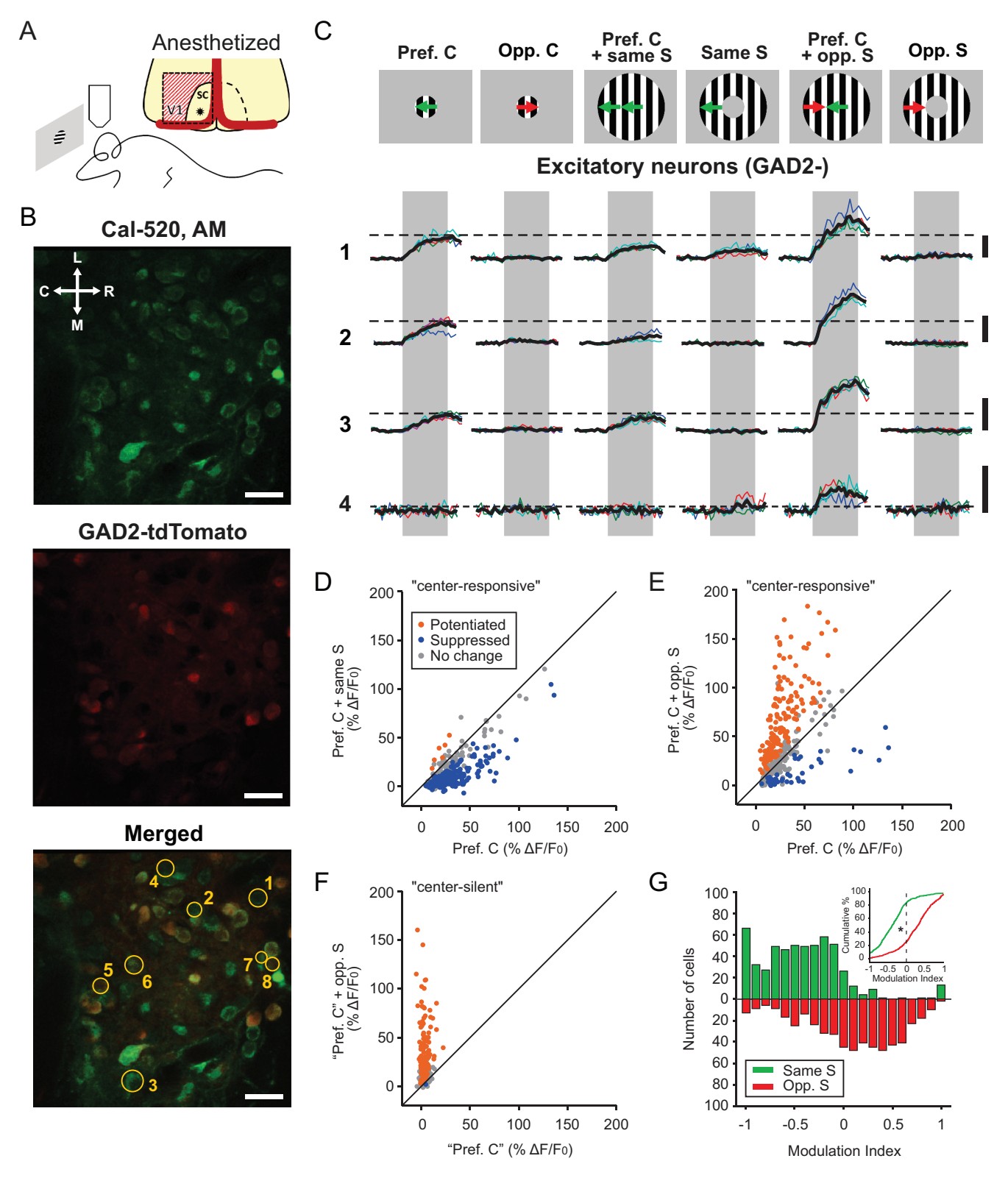

**Figure 1.** Excitatory neurons in the sSGS are bidirectionally modulated by motion direction in the receptive field surround. (**A**) Two-photon calcium imaging in the mouse sSGS (bottom). Depiction of the surgical procedure to expose the SC, showing the removal of V1 (top). The star indicates a rough estimate of the imaging location. (**B**) Field of view containing sSGS neurons (at 20 µm below the surface) loaded with Cal-520 (top), GAD2[+] neurons (expressing tdTomato) and GAD2[-] neurons (middle), and a merged image of both channels (bottom). Scale bars are 20 µm. R, rostral; C,

*Figure 1 continued on next page*

*Figure 1 continued*

caudal; M, medial; L, lateral. **(C)** Calcium signal of 4 GAD2⁻ neurons in response to six chosen conditions of the center-surround (C-S) stimulus. C, center; S, surround; Pref, preferred; Opp, opposite. The diagrams on top are for illustration purpose only, while the actual preferred directions vary from cell to cell. The numbers on the left represent the neurons circled in (**B**, bottom). Neurons 5, 6, 7, and 8 are GAD2⁺, and their responses are shown in *Figure 3A*. Thin multicolored traces are individual trials, and thick black traces are the average. All scale bars represent 100% Δ*F/F₀*. The dotted horizontal lines are aligned to the peak of the black trace in response to the preferred direction at the center. The gray boxes delimit the 2 s period of stimulus presentation. **(D–E)** Response comparison for individual center-responsive GAD2⁻ neurons at the preferred center direction and when the preferred center was coupled with the same-direction surround (**D**), or when coupled with opposite-direction surround (**E**, n = 355 cells, 9 mice). **(F)** Same plot as (**E**), but for center-silent neurons. See Results and Materials and methods for the determination of the 'preferred center direction' for these neurons (n = 191 cells, 9 mice). **(G)** Modulation index distribution under same-surround (green) and opposite-surround (red) conditions (n = 355 + 191=546 cells, 9 mice). Both histograms and cumulative distributions (inset) are shown. The color scheme used in panels (**D–F**) illustrates the results of a bootstrapping test to determine the significance of the C-S modulation for individual neurons (orange indicates potentiation; blue, suppression; gray, no statistically significant change; See Materials and methods for details).

DOI: https://doi.org/10.7554/eLife.35261.002

The following figure supplements are available for figure 1:

**Figure supplement 1.** Example receptive fields of GAD2⁻ and GAD2⁺ neurons in the sSGS.

DOI: https://doi.org/10.7554/eLife.35261.003

**Figure supplement 2.** Receptive field properties of GAD2⁻ and GAD2⁺ neurons in the sSGS.

DOI: https://doi.org/10.7554/eLife.35261.004

For most responsive excitatory neurons (n = 294/355, 82.8%), the small patch of gratings covered their entire RFs (*Figure 1—figure supplement 2A*), such that gratings in the surround (an annulus from 10 to 30° radius) did not cause any response when presented alone (e.g., *Figure 1C*). However, when the surround gratings were presented simultaneously with the center stimulus, the response magnitude of sSGS neurons was dramatically altered. In the case when both the center and surround gratings moved along the preferred direction of a given excitatory neuron, we saw a classical surround-suppression of the center response (*Figure 1C–D*). In striking contrast, when the preferred direction in the center was coupled with a surround stimulus of opposite direction, most excitatory sSGS neurons increased their responses (*Figure 1C,E*). In other words, to the same stimulus in their RF center, excitatory sSGS neurons could increase or decrease their response, that is, they are bidirectionally modulated, depending on what is shown in the surround.

Interestingly, a substantial population of excitatory sSGS neurons did not respond to the center or the surround gratings when presented separately but became responsive to particular Center-Surround (C-S) combinations (n = 191/811, 23.6%). This was despite the fact that the majority of these cells (154/191, 80.6%) had mappable RFs that were covered by the center patch (*Figure 1—figure supplement 1A,B* and *Figure 1—figure supplement 2C*). We assigned the 'preferred direction' of these 'center-silent' neurons as the center direction of their preferred C-S stimulus. When this 'preferred direction' in the center was coupled with its opposite direction surround, an emergent response was observed in those cells (e.g., cell four in *Figure 1C*, and *Figure 1F*).

To quantify the modulating effect of the surround, we calculated a "modulation index" for each neuron to compare its response to the center stimulus alone at the preferred direction ("preferred center", $R_{\text{pref. C}}$) with that to "preferred center" coupled with a particular surround ($R_{\text{pref. C with S}}$). The index, $\text{Modulation Index} = \frac{R_{\text{pref. C with S}} - R_{\text{pref. C}}}{R_{\text{pref. C with S}} + R_{\text{pref. C}}}$, ranged between -1 and 1 where negative values represented decreases in response and positive values represented increases. Over the entire population of these excitatory sSGS cells (n = 355 "center responsive" + 191 "center silent" = 546), the response to the "preferred center" stimulus became smaller with the introduction of the same direction surround in 83.3% of cells (n = 455/546), and larger in 74.7% of cells (n = 408/546) when the surround was moving in the opposite direction (Fig. 1G, Kolmogorov Smirnov [KS] test, p = 4.0e-83, KS stat = 0.59).

## sSGS excitatory neurons encode motion contrast

The above results demonstrate that excitatory neurons in the sSGS detect motion saliency by virtue of their sensitivity to the difference in direction between the RF center and surround. To study how sSGS neurons are tuned to this form of motion contrast, we systematically and independently varied

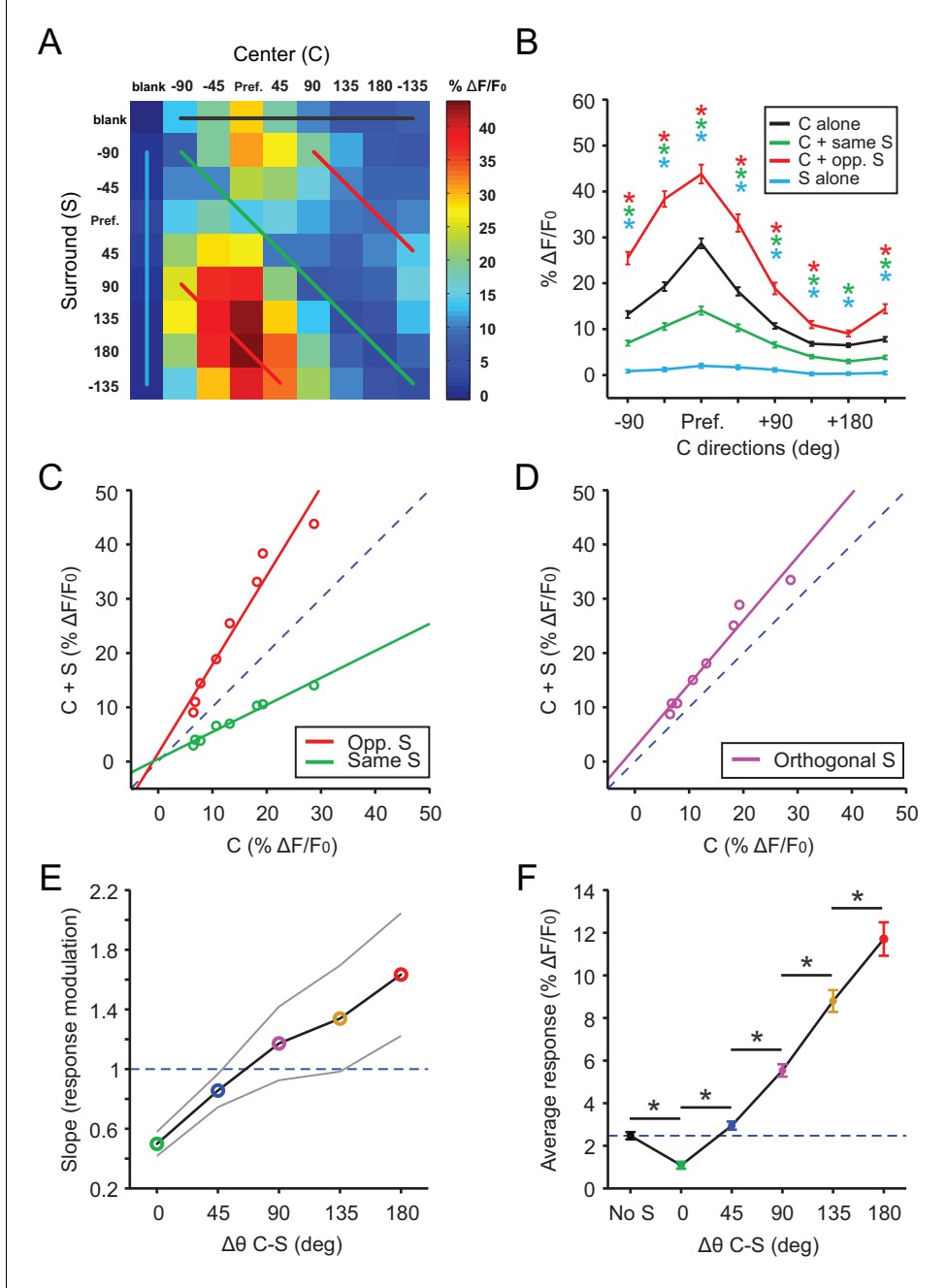

**Figure 2.** Excitatory neurons in the sSGS encode direction contrast. (**A**) Averaged response matrix of center-responsive GAD2⁻ neurons to all 81 combinations of the C-S stimulus, aligned to each cell's preferred direction (n = 355 cells, 9 mice). The color scale to the right represents the response magnitude in % $\Delta F/F_0$. (**B**) Aligned and averaged population tuning curves for these neurons under particular C-S combinations. The x-axis represents the direction of the center stimulus relative to the preferred direction ('Pref.'). The different colored curves represent the relationship of the surround to the center, corresponding to the same colored lines in (**A**). All data points are compared statistically to their corresponding points in the black tuning curve. (**C**) Geometric modulation of the center tuning curve by the two different surrounds in **B**; same surround induced divisive suppression (green, slope = 0.50, y-intercept = 0.48, $R^2$ = 0.97), and opposite surround induced multiplicative potentiation (red, slope = 1.63, y-intercept = 1.55, $R^2$ = 0.94). (**D**) Multiplicative potentiation of the center tuning curve by orthogonal-direction surrounds (slope = 1.17, y-intercept = 2.54, $R^2$ = 0.96). The dashed blued lines in **C** and **D** are lines of identity. (**E**) The slopes of modulation illustrated in (**C-D**, corresponding colors) as well as the intermediate conditions vs. C-S direction difference (gray lines delimit the 95% confidence interval). The dashed blued line

*Figure 2 continued on next page*

*Figure 2 continued*

indicates a slope of 1, that is, no modulation. (F) Mean averaged responses ($\Delta F/F_0$) of center-silent GAD2⁻ neurons vs. C-S direction difference (n = 191 cells, 9 mice). The dashed blue line is averaged 'response' to center alone. Data in **B** and **F** are presented as mean ± s.e.m. *: p<0.05, Mann-Whitney *U*-test.

DOI: https://doi.org/10.7554/eLife.35261.005

The following figure supplement is available for figure 2:

**Figure supplement 1.** Center-surround ('C-S') interactions in GAD2⁻ neurons.

DOI: https://doi.org/10.7554/eLife.35261.006

the direction of the center and surround of the C-S gratings. Specifically, our stimulus set consisted of 81 C-S combinations (*Figure 2A*; 8 directions and one blank for both center and surround).

We shifted the 81 condition response matrix of all excitatory cells that responded to center gratings (n = 355) in order to align their preferred directions. These shifted responses were then averaged and illustrated in a 9 × 9 matrix (*Figure 2A*). This matrix allows us to examine the responses under different C-S conditions. For example, the top row (Black trace in *Figure 2A*) represents the stimulus condition where the surround was 'blank', that is, only center grating was shown. As expected, these neurons were DS in response to the center grating alone, showing greater responses to the preferred directions than to the opposite direction (Black trace in *Figure 2A–B*). Furthermore, surround suppression was seen for all directions of the center stimulus when the surround gratings were moving in the same direction as the center (i.e., the diagonal in the matrix; Green traces in *Figure 2A–B*). Similarly, the potentiation by opposite surround was also seen for all directions of the center stimulus (Red traces in *Figure 2A–B*).

Interestingly, this modulation was geometric in nature, ranging from a divisive suppression by the same surround to a multiplicative potentiation by the opposite surround (*Figure 2C*). In fact, a linear modulation of the center responses was also seen with intermediate differences between C-S directions (e.g., *Figure 2D*). We thus calculated the slope of these linear relationships (i.e., fold changes of center responses by the surround), and found that it gradually increased with the C-S direction difference (*Figure 2E*). In other words, excitatory sSGS neurons were monotonically tuned to motion contrast, showing maximal responses to the most salient stimulus with oppositely-moving center and surround. We also performed similar analyses with normalized responses for each cell before averaging and reached the same conclusions (data not shown). Furthermore, in addition to analyzing the averaged data, which reveal how sSGS excitatory neurons encode motion contrast as a population, we also examined the geometric relationship in individual cells. There was great variability in the goodness of fit between Center alone and C-S responses (*Figure 2—figure supplement 1A–D*), but the vast majority of them could be reasonably fitted by a linear relationship ($R^2 \geq 0.5$) with near zero y-intercept (*Figure 2—figure supplement 1E*), suggesting a multiplicative modulation. Importantly, the average slope increased monotonically with the C-S direction difference (*Figure 2—figure supplement 1F–I*), just like for the population response.

Finally, we performed a similar analysis for the 'center silent' neurons (n = 191), by averaging their responses to particular C-S direction differences. An emergent, and again monotonically increasing, response was seen as the C-S direction difference increased (*Figure 2F*). These cells thus display similar tuning to direction contrast as their center-responsive neighbors.

## Inhibitory neurons in the sSGS are suppressed by motion contrast

Next, we analyzed how inhibitory sSGS neurons (GAD2⁺) responded to C-S direction contrast. Just like excitatory neurons, inhibitory neurons also experienced surround suppression when the surround direction matched the direction in the RF center (*Figure 3A–B*). They were, however, significantly less suppressed than their excitatory counterparts (*Figure 4A*, KS test, p=6.0e-20, KS stat = 0.30). Furthermore, when the surround grating drifted in the opposite direction, the inhibitory neurons' response to the preferred center stimulus was quite strongly suppressed (*Figure 3A and C*). This is in stark contrast with the potentiation seen in excitatory sSGS neurons under the same conditions (*Figure 4B*, KS test, p=1.46e-58, KS stat = 0.51). Consistently, compared with excitatory neurons, fewer 'center-silent' inhibitory cells became responsive when opposite center and surround was shown and their responses were weaker (n = 85/652; *Figure 3D*, compare with *Figure 1F*). Overall,

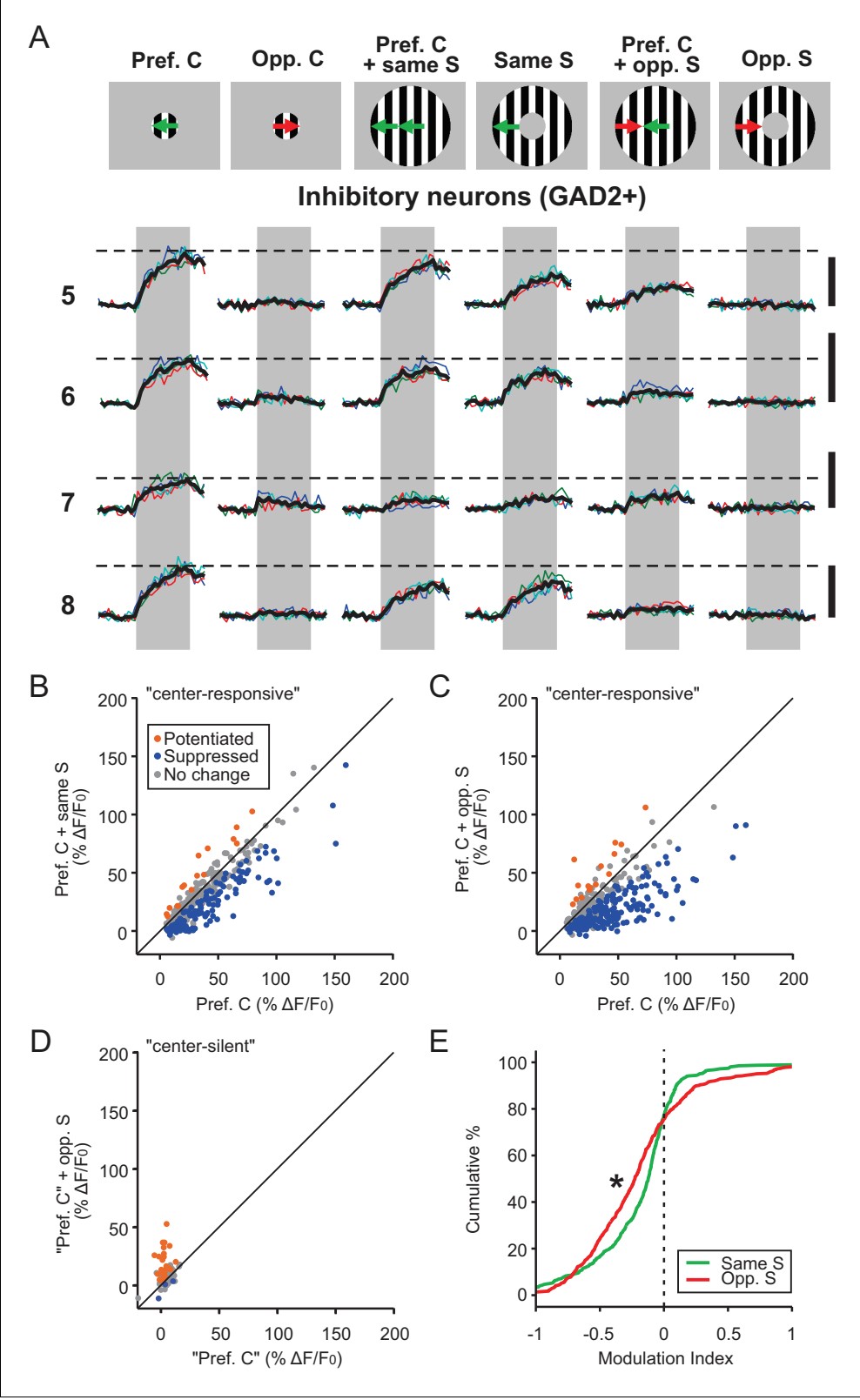

**Figure 3.** Inhibitory neurons in the sSGS are suppressed by direction contrast. (**A**) Same as in *Figure 1C*, for four inhibitory neurons, with numbers on the left representing the neurons circled in *Figure 1B*, bottom. (**B–C**) Response comparison for individual center-responsive GAD2+ neurons to the preferred center direction and when the preferred center was coupled with same-direction surround (**B**), or opposite-direction surround (**C**, n = 379, 9

*Figure 3 continued on next page*

*Figure 3 continued*

mice). The color scheme follows that in *Figure 1D–F*. (D) Response comparison for GAD2⁺ neurons that were silent to the separate presentations of center and surround but responded to a C-S combination, at the 'preferred center' and when coupled with the opposite-direction surround. See Materials and methods for the determination of the 'preferred center' for those neurons (n = 85 cells, 9 mice). (E) Modulation index distribution of neurons in (B–D) under same-surround (green) and opposite-surround (red) conditions (n = 379 + 85 = 464 cells, nine mice, KS test, p = 3.2e-6, KS stat = 0.17).

DOI: https://doi.org/10.7554/eLife.35261.007

The following figure supplement is available for figure 3:

**Figure supplement 1.** Center-surround interactions in GAD2⁺ neurons.

DOI: https://doi.org/10.7554/eLife.35261.008

the suppression by the opposite surround was even greater than that by the same-direction surround for most inhibitory neurons (*Figure 3E*, KS test, p=3.2e-6, KS stat = 0.17), as well as at the population level (compare green and red curves in *Figure 3—figure supplement 1A–B*).

In addition to calculating a modulation index, we also used a bootstrapping test to determine statistical significance for individual neurons when comparing their response to C-S combinations with that to the center stimulus alone (see Materials and methods for details). Consistently, using this method, we found that a much larger proportion of excitatory sSGS neurons was significantly potentiated by the opposite-direction surround (n = 274 out of 546 responsive cells, 50.2%), compared to inhibitory neurons (n = 46 out of 464, 9.9%; *Figure 4C*).

Finally, we found that many center-responsive inhibitory neurons (204/379, 53.8%) could be activated by a surround alone that was moving in their preferred direction (*Figure 3A* and *Figure 3—figure supplement 1B*, blue curve). Such a response profile was rarely observed in excitatory neurons. This was quite surprising given the fact that excitatory and inhibitory neurons had similar RF sizes when mapped with flashing squares (*Figure 1—figure supplement 1* and *Figure 1—figure supplement 2A–F*, p=0.19, Mann-Whitney *U*-test comparing their RF size in panels E-F, n = 542 GAD2⁻ and 425 GAD2⁺ neurons responsive to flashing squares). It is possible that flashing squares might not provide enough drive to activate inhibitory neurons away from the center, leading to an

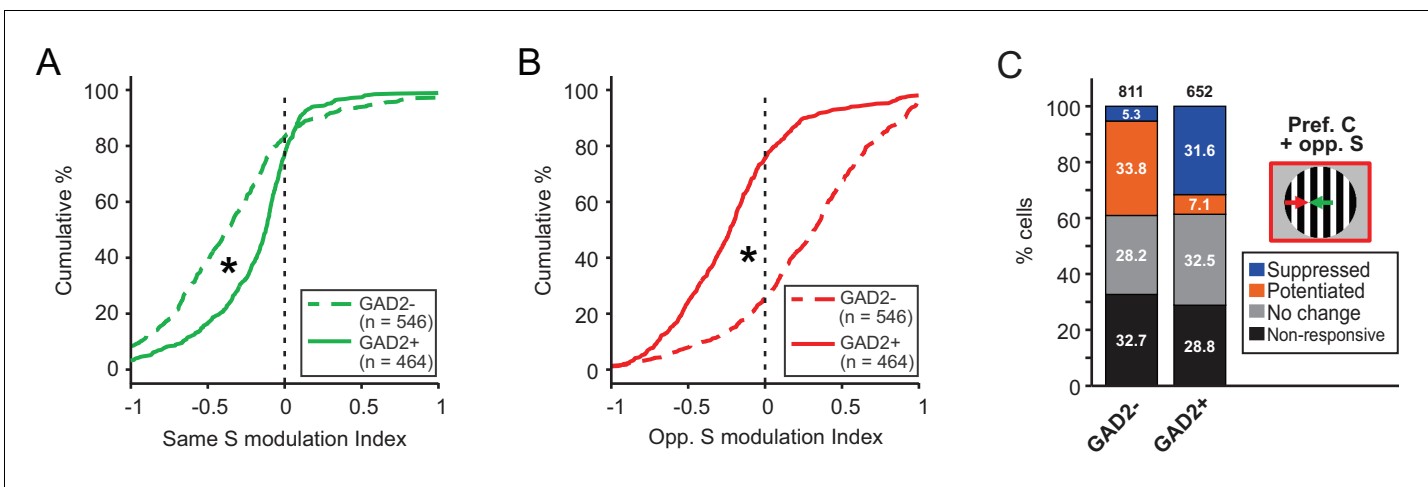

**Figure 4.** Excitatory and inhibitory sSGS neurons are differently modulated by the surround. (A–B) Cumulative distributions of modulation index by same surround (A) or opposite surround (B) in GAD2⁻ (dashed lines) and GAD2⁺ cells (solid lines). The distributions are the same as in *Figure 1G* and *Figure 3E*, grouped here to highlight the differences between the two cell types. (C) Percentages of GAD2⁻ and GAD2⁺ neurons in four response categories (determined by a bootstrapping test) to the presentation of the preferred center + opposite surround combination: non-responsive, non-modulated, potentiated, and suppressed. Values in the boxes represent the percentage of neurons in each category; numbers at the top represent the total numbers of neurons in the study. The 'non-responsive' category included neurons that did not respond to any of the C-S conditions and a small population of neurons that responded to the surround alone, but not the center stimulus.

DOI: https://doi.org/10.7554/eLife.35261.009

underestimation of their effective RF size. Drifting gratings, on the other hand, could provide that drive, thereby revealing a potential difference between the RF properties of excitatory and inhibitory neurons in the sSGS. Alternatively, but non-exclusively, a wider spread of GAD2$^+$ RFs compared to their GAD2$^-$ neighbors could lead GAD2$^+$ neurons to sample inputs from beyond the center drifting gratings stimulus, therefore explaining those observations. This was true to some extent. The 2-dimensional standard deviation of GAD2$^+$ RF centroids for any given imaging field of view was statistically larger than that of GAD2$^-$ ones (*Figure 1—figure supplement 2G and H*, p=0.03, paired t-test). However, while this difference is statistically significant, it is rather subtle, and unlikely to account for the surround-evoked responses in GAD2$^+$ cells.

Regardless of their origin, we performed two additional analyses to determine whether the 'surround-alone' responses could confound our conclusions. In one, we compared the responses of these cells to C-S stimuli vs. the sum of their respective responses to Center alone and to Surround alone (*Figure 3—figure supplement 1C–D*). In the other, we only included the inhibitory cells that did not show any significant response to the Surround alone stimulus and examined their response profiles (n = 175/379, *Figure 3—figure supplement 1E*). Both analyses support the conclusion that inhibitory cells were suppressed by a surround stimulus, regardless of its direction. Together, these results demonstrate that sSGS excitatory and inhibitory neurons have slightly different RF properties and are differentially modulated by motion contrast. The greater suppression of inhibitory neurons by the opposite-direction surround suggests a possible role for them in mediating the potentiation of excitatory neurons through disinhibition.

## sSGS are specifically tuned to motion direction contrast

The responses we observed in the sSGS could be specific to the particular property of motion direction. Alternatively, such responses could be elicited by any type of feature-contrast between center and surround, in an indiscriminate manner. To assess the specificity of these responses, we compared the response of sSGS neurons to several types of C-S feature contrasts, including direction, phase, temporal frequency, and static orientation differences.

We first presented the mice with a surround stimulus drifting in the same direction as the center, but 180° out of phase ('Anti-phase S', *Figure 5A*). Compared to the in-phase surround ('Same S'), which strongly suppressed sSGS responses to the center stimulus, the anti-phase surround resulted in an attenuated suppression (or even a slight potentiation) in both GAD2$^-$ and GAD2$^+$ neurons (*Figure 5B–C*; compare the green and the red dashed lines in *Figure 5B*, p=2.2e-10, and *Figure 5C*, p=6.2e-6). Similar responses were seen when we varied the temporal frequency of the surround grating. Either lower (1 Hz, red dotted) or higher (4 Hz, navy blue dotted) temporal frequencies in the surround elicited a similar attenuation of response suppression compared to a uniform (2 Hz, solid green) temporal frequency between center and surround (compare the solid green line to the dotted red and blue lines in *Figure 5B–C*, p<0.01 for both comparisons).

Importantly, the attenuated suppression seen in these 3 sets of center-surround contrast (anti-phase, and lower and higher temporal frequency) did not reach the level of potentiation experienced by the same excitatory neurons to an oppositely-moving surround (solid red line in *Figure 5B*, n = 73, 3 mice, KS test, p<0.01 for all comparisons). In the same vein, the heightened level of suppression experienced by inhibitory neurons to an oppositely-moving surround could not be matched either under the aforementioned conditions (solid red line in *Figure 5C*, n = 124, 3 mice, KS test, p<0.01 for all comparisons). In fact, as mentioned earlier, and as a point of major divergence, inhibitory neurons experienced a significant alleviation of suppression under those conditions, when compared to non-contrasting C-S stimulus conditions (*Figure 5C*).

Next, we tested static gratings with varying orientations. Consistent with their selectivity for the direction of moving stimuli, sSGS neurons showed weaker responses to static gratings (data not shown). More relevant to the current study, both GAD2$^-$ and GAD2$^+$ neurons experienced a slightly attenuated level of suppression under cross-surround conditions, but no potentiation, compared to iso-surround (*Figure 5D–E*). In addition, the excitatory neurons (i.e., GAD2$^-$) appeared to be more strongly suppressed than the inhibitory neurons (i.e., GAD2$^+$) regardless of the surround orientation (Compare same-colored distributions in *Figure 5D and E*; Iso-surround GAD2$^-$ vs. GAD2$^+$, KS test, p=1.6e-8, KS stat = 0.37; Cross-surround GAD2$^-$ vs. GAD2$^+$, KS test, p=9.7e-7, KS stat = 0.33). In other words, the bidirectional modulation of activity that we observed in GAD2$^-$ vs. GAD2$^+$ neurons

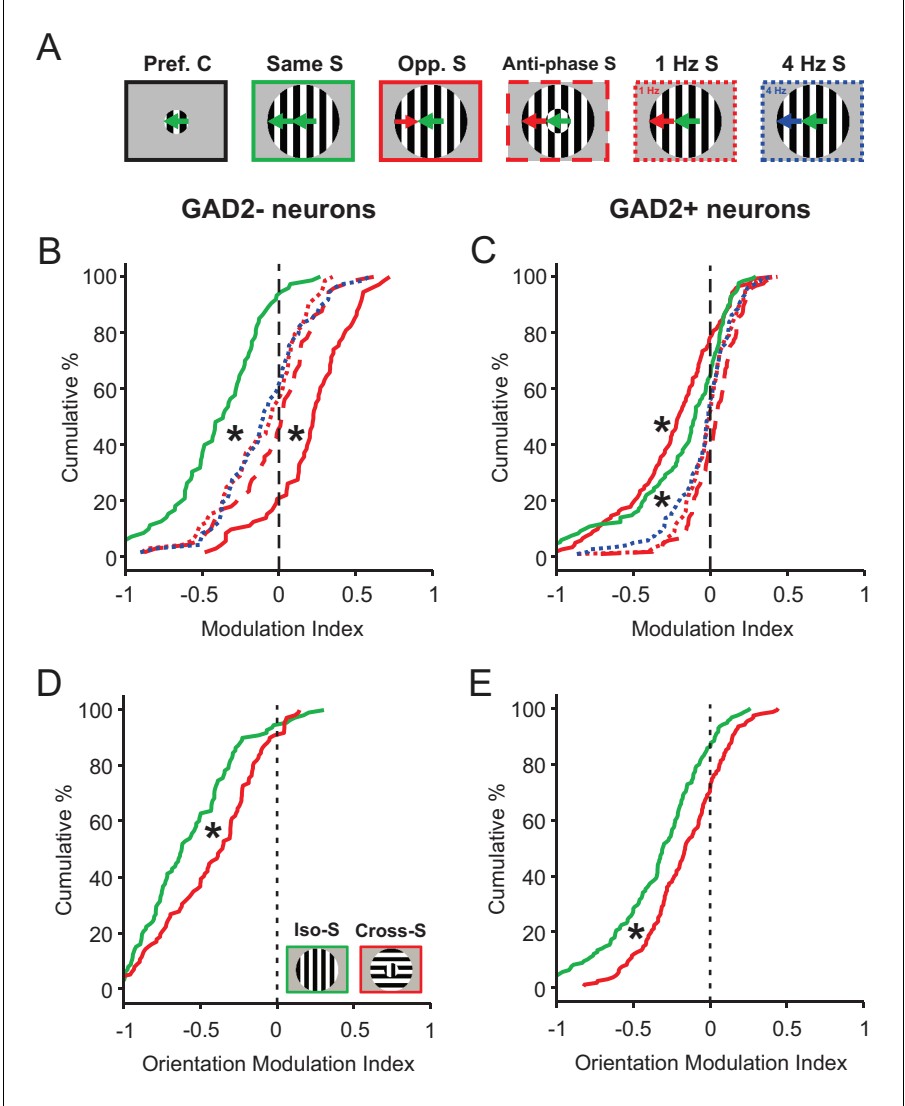

**Figure 5.** Phase, temporal frequency, and orientation contrasts have a different modulatory effect on sSGS neurons compared to direction contrast. (**A**) Visual stimuli used in this set of experiments and analyses, with center gratings presented at individual cells' preferred directions, either alone (Pref. C, left) or surrounded by different patterns of drifting grating: surround along the same direction, opposite direction, anti-phase along the same direction, and different temporal frequencies (Opp., Opposite; S, Surround). (**B–C**) Modulation index quantifying how the response to each C-S stimuli differ from that to center grating only. The same color and line styles follow those in **A**. Plot **B** is for center-responsive GAD2⁻ neurons (n = 73, 3 mice, KS test, p<0.01 between opposite direction surround and any of the three dotted or dashed lines; and p<0.01 between same direction surround and any of the three dotted or dashed lines). Plot **C** is for GAD2⁺ neurons (n = 124, 3 mice, KS test, p<0.01 between opposite direction surround and any of the three dotted or dashed lines, and p<0.01 between same direction surround and any of the three dotted or dashed lines). (**D–E**) Modulation index for responses to static oriented gratings in center and surround, under iso-surround (green) and cross-surround (red) conditions in center-responsive GAD2⁻ (**D**, n = 111, 3 mice, KS test, p = 2.5e-4, KS stat = 0.28), and GAD2⁺ neurons (**E**, n = 155, 3 mice, KS test, p = 5.8e-4, KS stat = 0.23).

DOI: https://doi.org/10.7554/eLife.35261.010

under C-S motion direction contrast was not observed under static orientation opponency conditions.

Altogether, these observations reveal a specificity of saliency responses in sSGS neurons to motion direction contrast between the center and surround. The same cells respond both

qualitatively and quantitatively differently when the features that render the center stimulus salient are related to other aspects of motion or to static orientation.

## Depth-dependent motion contrast coding in the SGS

Studies in a number of species have shown that the visual layers of the SC or optic tectum can be further divided into sub-laminae (*May, 2006*). Indeed, based on a small number of single unit recordings, we recently found that direction selectivity in the mouse SGS declines with depth (*Inayat et al., 2015*). We thus assessed the depth profile of motion direction contrast responses and its relationship to direction selectivity. We were limited in our imaging depth when using calcium indicators that disperse throughout the cell body and processes, due to the strong neuropil signal in the deeper SGS. To overcome this limitation, we used a genetically-encoded calcium indicator (AAV-H2B-GCaMP6s) that was largely restricted to the cell nucleus (*Figure 6A*). This led to a substantial reduction of the neuropil signal in the deeper layers of the SGS, and allowed us to confidently image down to depths of around 200 µm. We characterized the performance of this indicator by performing simultaneous two-photon imaging and cell-attached recording to correlate the fluorescent signal with spiking activity. Although the 'nuclear' GCaMP6s was significantly slower than Cal-520 and could not resolve single spike activity (*Figure 6—figure supplement 1A–C*), it was able to reliably report the tuning of SGS neurons to the C-S stimuli (*Figure 6—figure supplement 1D–E* and *Figure 6—figure supplement 2*).

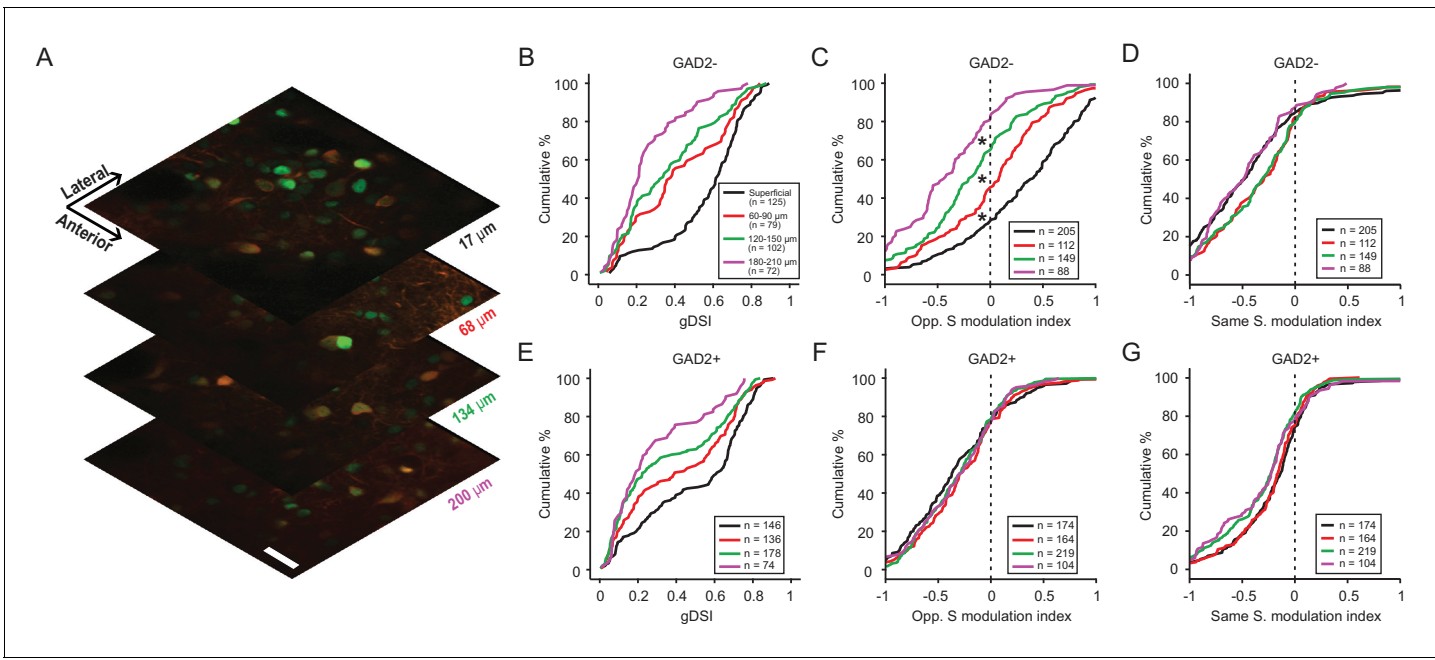

**Figure 6.** Direction contrast sensitivity declines with depth in the SGS. (**A**) Two-photon calcium imaging at different depths of the SGS, using AAV-H2B-GCaMP6s. Shown are neurons expressing H2B-GCaMP6s at four different depths in the SGS of a GAD2-tdTomato mouse. Scale bar is 20 µm. (**B**) Cumulative distribution of gDSI divided into four depth categories for center-responsive GAD2⁻ neurons (n = 378, 10 mice). The same depth color code applies to panels **B-E**. (**C**) Cumulative distribution of the opposite-surround modulation index for center-responsive and center-silent GAD2⁻ neurons (n = 378 + 176=554, 10 mice). (**D**) Cumulative distribution of the same-surround modulation index for GAD2⁻ neurons (n = 554, 10 mice; p<0.05 between 'superficial' and 60 µm, and between 120 µm and 180 µm, KS test). (**E–G**) Same as in (**B–D**), but for GAD2⁺ neurons (n = 534 in D; and n = 534 + 127=661 in E, 10 mice; In **G**, p<0.05 between 60 µm and 120 µm, KS test).

DOI: https://doi.org/10.7554/eLife.35261.011

The following figure supplements are available for figure 6:

**Figure supplement 1.** Characterization of H2B-GCaMP6s activity with cell-attached recording.

DOI: https://doi.org/10.7554/eLife.35261.012

**Figure supplement 2.** Varying the time window of H2B-GCaMP6s signal analysis does not impact the main findings.

DOI: https://doi.org/10.7554/eLife.35261.013

We first quantified SGS neurons' direction selectivity in response to the center gratings. Largely consistent with the results using Cal-520, the very superficial SGS lamina was enriched with highly DS cells, including both excitatory and inhibitory neurons (black curves in Figure 6B and E, respectively; 109/125, 87.2% of GAD2$^-$ and 106/146, 72.6% of GAD2$^+$ neurons had gDSI $\geq$0.25). The degree of direction selectivity declined with depth in the SGS, confirming our previous single unit results and results from a recent study using high-density electrode recordings (*Ito et al., 2017*). Importantly, our results indicate that the decline was observed for both excitatory (*Figure 6B*, KS test, p=1.7e-16, KS stat = 0.62, between the most superficial (black) and deepest (magenta) cell populations) and inhibitory neurons (*Figure 6E*, KS test, p=4.6e-7, KS stat = 0.39).

We then examined the depth profile of the surround modulation. In response to oppositely-moving surround, the response potentiation of excitatory neurons that we observed in the sSGS using Cal-520 was confirmed with nuclear GCaMP6s. Interestingly, this potentiation gradually turned into suppression with depth (*Figure 6C*, KS test, p<0.01 between the four depths). On the other hand, surround of the same direction remained suppressive across depth, despite some subtle difference (*Figure 6D*). In the case of inhibitory neurons, no significant change was seen in the opposite surround modulation index with depth (*Figure 6F*) and only minor difference in the same surround modulation index (*Figure 6G*), but overall the cells remained similarly suppressed by the opposite or same surround.

The concurrent decline in excitatory cells' gDSI and modulation index with depth suggested a potential correlation between these properties. Indeed, a significant, albeit noisy, correlation was seen between the two variables for excitatory neurons (*Figure 7A*, r = 0.38, p=1.3e-14), where highly DS neurons tend to be potentiated by the opposite surround, while the non-selective ones tend to be suppressed. In contrast, we observed a negative correlation for inhibitory neurons between the modulation index and gDSI (*Figure 7B*, r = −0.42, p=4.2e-24), where the highly DS cells were much more suppressed by the opposite surround. Interestingly, a positive correlation was seen between the modulation index and gDSI for inhibitory neurons when the same surround was presented (*Figure 7C*, r = 0.43, p=8.1e-26). Correlations of the same polarities were also observed for the sSGS excitatory and inhibitory neurons that were imaged with Cal-520 (*Figure 7—figure supplement 1*).

Imaging deeper into the SGS also allowed us to reveal an intriguing property of inhibitory neurons. These cells appear to form two clusters based on their gDSI (*Figure 7B–C*). The more DS cells

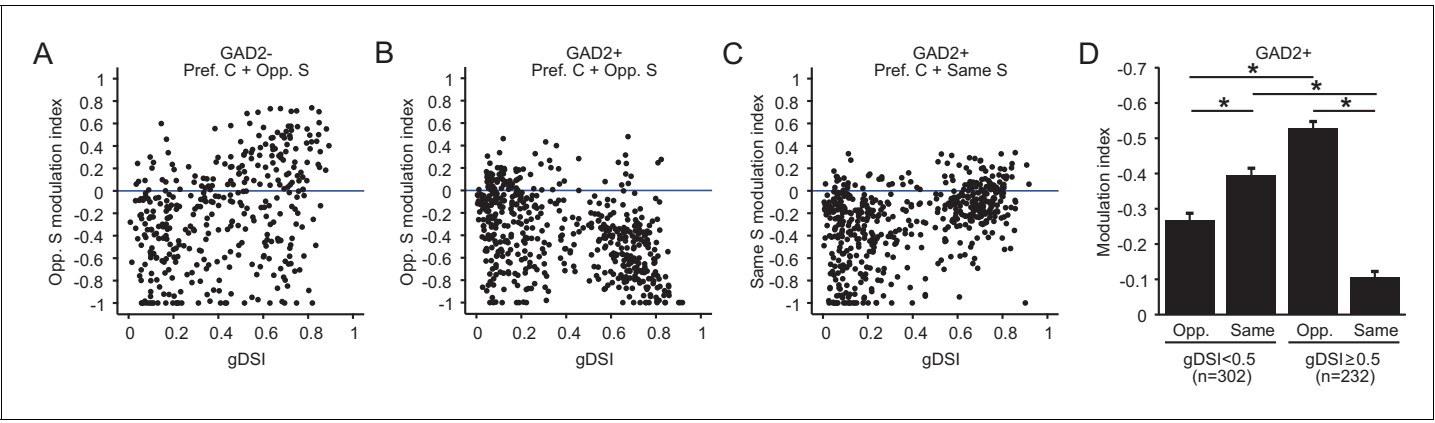

**Figure 7.** Relationship between surround modulation and direction selectivity. (A–B) Relationship between the opposite-surround modulation index and gDSI for center-responsive GAD2$^-$ neurons (A, n = 378, 10 mice) and GAD2$^+$ neurons (B, n = 534, 10 mice) at all depths combined. (C) Relationship between the same-surround modulation index and gDSI for the same cells in B. (D) Modulation index by same or opposite surround for GAD2$^+$ neurons, separated into two gDSI categories (gDSI < 0.5, n = 302; gDSI $\geq$ 0.5, n = 232, 10 mice. Data are presented as mean ± s.e.m. * represents p<0.01, Mann-Whitney U-test).

DOI: https://doi.org/10.7554/eLife.35261.014

The following figure supplement is available for figure 7:

**Figure supplement 1.** Relationship between surround modulation and direction selectivity of sSGS neurons imaged by Cal-520.
DOI: https://doi.org/10.7554/eLife.35261.015

(gDSI $\geq$ 0.5) were much more susceptible to suppression by the opposite direction surround compared to less selective counterparts (*Figure 7B*, KS test, p=1.1e-19, KS stat = 0.41, between the 'gDSI $\geq$ 0.5' and 'gDSI < 0.5' populations; *Figure 7D*, Mann-Whitney U-test, p=3.7e-19, between 'gDSI $\geq$0.5 Opp.' and 'gDSI < 0.5 Opp.''), and much less susceptible to suppression by the same surround (*Figure 7C*, KS test, p=2.5e-19, KS stat = 0.40, between the 'gDSI $\geq$ 0.5' and 'gDSI < 0.5' populations; *Figure 7D*, Mann-Whitney U-test, p=7.5e-24, between 'gDSI $\geq$ 0.5 Same' and 'gDSI < 0.5 Same'). Whether they correspond to different functional classes of inhibitory neurons and how they might be involved in motion contrast coding remain to be determined in future studies.

## Discussion

In this study, we determined how neurons in the mouse SGS encode motion contrast between their RF center and surround. The responses of superficial excitatory neurons are bidirectionally modulated, increasing monotonically as a function of the direction difference between the center and surround, from suppression by the same-direction surround to maximal potentiation by an oppositely-moving surround. Such response profiles are likely important for the animal to detect object motion in the environment and distinguish it from self-induced global motion in the background.

### Saliency computation and representation

Current theories postulate that visual saliency is analyzed separately by feature-specific channels, which are then combined into a feature-agnostic saliency map (*Veale et al., 2017*). A classic example of feature-specific saliency computation takes place in V1 (*Li, 2002*). Being orientation selective, V1 neurons modulate their responses depending on the orientation difference between the RF and its surround. In primate and cat V1, lower levels of suppression, or even facilitation, could occur when cross-oriented stimuli were shown in the surround (*Jones et al., 2001*; *Jones et al., 2002*; *Kastner et al., 1999*; *Knierim and van Essen, 1992*; *Nothdurft et al., 1999*; *Sengpiel et al., 1997*; *Sillito et al., 1995*). More recent studies have shown similar findings in mouse V1, where responses were suppressed by an iso-oriented surround, but experienced an attenuation of suppression to a cross-oriented surround (*Self et al., 2014*). This type of differential neuronal activity may underlie the 'pop-out' phenomenon mentioned earlier (*Li, 1999*). Similarly, the direction-contrast dependent response might help the animal distinguish between self-induced motion in the visual scene, manifested as full-field motion, and actual object motion in the RF. Indeed, a similar role has been proposed for a population of neurons in mouse V1, under awake and running conditions (*Keller et al., 2012*; *Zmarz and Keller, 2016*). Importantly, mouse studies have allowed researchers to explore the circuit mechanisms underlying surround suppression in more detail. With the available genetic toolkit in this species, the specific contributions of different types of cortical inhibitory neurons to surround-suppression are now being elucidated (*Adesnik et al., 2012*; *Nienborg et al., 2013*).

In lower vertebrates where neocortex has not evolved, the SC homologue optic tectum is the main visual center for signal processing, including saliency analysis. Tectal neurons in these species are usually motion sensitive and selective for movement direction. In barn owls, tectal neurons are differentially suppressed depending on the motion direction in the surround (*Zahar et al., 2012*). In the pigeon tectum, a potentiating effect could be elicited under conditions of center-surround motion-opponency (*Frost et al., 1981*; *Sun et al., 2002*). Additionally, studies in the archer fish showed that their tectal neurons exhibited contextual modulation which might underlie 'pop-out' in a visual search paradigm (*Ben-Tov et al., 2015*).

In the primate SC, very few visual neurons are tuned to specific features such as direction or orientation. The SC is therefore thought to be the locus of integration of feature-specific cortical inputs into a feature-agnostic saliency map (*Veale et al., 2017*), where neurons would respond indiscriminately to any feature contrast between the RF center and the surround. In contrast, neurons in the mouse SGS are mostly tuned to particular visual features such as motion direction. We demonstrate here that sSGS neurons in fact perform feature-specific saliency computations by encoding direction contrast in a monotonic, bidirectional, and cell-type specific fashion. Our findings are largely consistent with previous studies of orientation contextual modulation in rat SC (*Girman and Lund, 2007*), and more recently in mouse SC (*Ahmadlou et al., 2017*), where SGS neurons show greater responses when the center grating is surrounded by cross-orientated surround than by iso-oriented

surround. Interestingly, we found that this computation is cell-type specific and more prominent in the very superficial SGS. In the mouse SC, direction selectivity is lost in the deeper laminae of the SGS, and the intermediate layers are multisensory integrators (*Cang and Feldheim, 2013*; *Inayat et al., 2015*). It is therefore possible that the transformation from a feature-specific saliency analysis to a feature-agnostic saliency map, or even more generally to a modality-agnostic saliency map, takes place between the deep SGS and the intermediate layers of the SC. Our results thus significantly expand on past findings of contextual modulation in the tectum as well as the SC. Importantly, the comparison of SC response properties across vertebrate species, including our current findings, supports the idea of a gradual evolutionary migration of saliency computation from a single locus in the SC/OT to a multi-structural process that involves cortical inputs (*Zhaoping, 2016*).

## Mechanisms for motion contrast computation in the mouse SGS

One of the main reasons for studying saliency computation in mice is that we can monitor the activity of specific cell types using modern genetic and imaging techniques. Here we show that SGS excitatory and inhibitory neurons respond differently to the same motion contrast stimuli, an important finding that had not been shown in any other species. The inhibitory circuits in the rodent SGS have only been studied in the context of classical surround suppression, mostly using stimulus size tuning as a measure of modulation (*Binns and Salt, 1997*; *Kasai and Isa, 2016*). A recent two-photon imaging study in the mouse SGS showed that the activity of local inhibitory and excitatory neurons is equally suppressed by the surround, implicating long range inhibitory input in mediating the phenomenon (*Kasai and Isa, 2016*). In our current study, however, we observe interesting differences in the responses of excitatory and inhibitory neurons. Inhibitory neurons are less susceptible to surround suppression, especially when they are direction selective (*Figure 7*). These neurons can nonetheless be much more suppressed by a surround moving in the opposite direction (*Figure 7*), potentially contributing to the increased responses in excitatory neurons.

Input from other brain areas could also contribute to surround modulation of SGS activity. The SGS receives direct inputs from both retina and visual cortex, in addition to a few other structures (*May, 2006*). Visual cortex was removed in our experiments, ruling out its involvement in setting up those responses. This is consistent with the findings of a recent electrophysiological study suggesting that visual cortex may actually limit context-dependent modulation by cross-oriented gratings in the SGS (*Ahmadlou et al., 2017*). The same study also showed that this type of contextual modulation was more prominent in awake mice than in anesthetized mice (*Ahmadlou et al., 2017*). How the direction-specific contextual modulation we discovered here might be influenced by anesthesia, and how it might be modulated by cortex in awake mice, remain to be determined.

Retinal inputs, on the other hand, were shown to be the source of direction selectivity in the SGS (*Shi et al., 2017*) and could provide contextually-modulated input. Surround-modulated suppression was observed under several motion-contrast regimes (spatial phase, spatial frequency, and velocity) in direction-selective retinal ganglion cells (DSGCs) in rabbits (*Chiao and Masland, 2003*), and orientation-specific surround modulation of RGCs was shown in rats (*Girman and Lund, 2010*). Furthermore, object motion sensitive ('OMS') RGCs were found in rabbits and salamander, which are suppressed by global motion but respond strongly to motion difference between RF center and surround (*Olveczky et al., 2003*; *Baccus et al., 2008*). The OMS cells included several RGC types, and their selectivity for differential motion is independent of direction (*Olveczky et al., 2003*). Even more relevantly, the same stimuli used in our experiments have also been used to investigate context modulation in mouse DSGCs in an unpublished study (Xiaolin Huang and Wei Wei, University of Chicago, personal communications). They showed that the oppositely-moving surround elicits smaller suppression than the same-direction surround, but never reaches potentiation; and the anti-phase surround causes similar responses as the opposite surround. These retinal studies thus suggest that some of the SC responses, including the suppression by same surround and its attenuation by anti-phase and cross-oriented static gratings, are likely inherited from the retina. However, these and other observations indicate that the bidirectional encoding of motion direction contrast by sSGS neurons is not inherited from the retina. Most notably, the modulation by out-of-phase and opposite surround is nearly identical in DSGCs, but dramatically different in the sSGS. Additionally, the fact that excitatory and inhibitory neurons in the sSGS, both of which receive direct retinal input (*Shi et al., 2017*), exhibit strikingly different responses to center-surround stimuli argues for a role of intracollicular circuit mechanisms. These circuits may include long range excitatory and inhibitory

connections from the surround, and they must be wired in a direction- and cell-type-specific manner to mediate the differential responses in excitatory and inhibitory neurons. They are also likely restricted to the superficial SGS to account for the depth-specific changes.

In conclusion, our study identified response correlates of motion saliency in the mouse SGS. The striking distinction between the responses of excitatory and inhibitory neurons to direction contrast in this structure makes it a strong candidate to be a locus of saliency encoding. This opens the door for future mechanistic studies that manipulate local inhibitory circuits in the SGS and examine the cellular and behavioral consequences. Our findings thus offer a unique opportunity to describe a circuit-level mechanism of saliency computation in the brain, and to look for downstream neuronal populations where feature-specific saliency representations are integrated into a feature-agnostic saliency map.

## Materials and methods

### Animal preparation

Adult C57BL/6 mice of both sexes were used in this study (n = 24, 2–4 months old), *Gad2*-IRES-cre mice (from the Jackson Laboratory, Stock no. 010802) were either crossed with an Ai9 line (*RCL-tdT*, Stock no. 007909), or injected with AAV1.CAG.Flex.tdTomato.WPRE.bGH (University of Pennsylvania Vector Core, Allen Institute 864) in their SC, to express the red fluorescent protein tdTomato in glutamate decarboxylase two positive (GAD2$^+$, GABAergic) neurons. All mice were kept on a 12 hr light/dark cycle, with one to five animals housed per cage. All experimental procedures were approved by the Northwestern University Institutional Animal Care and Use Committee.

Mice were anesthetized with urethane (1.2 g/kg in 0.9% saline solution, i.p.) and then sedated with chlorprothixene (10 mg/kg in water, i.p.). Atropine (0.3 mg/kg in 0.9% saline) and dexamethasone (2 mg/kg in 0.9% saline) were subsequently administered subcutaneously to minimize respiratory secretions and brain inflammation, respectively. The animals were then transferred onto a heating pad, and their body temperature was monitored via a rectal thermoprobe and maintained at 37°C through a feedback heater control module (Frederick Haer Company, Bowdoinham, Maine). Artificial tears (Henry Schein) were applied to the eyes for protection during surgery. The scalp was then shaved, and the skin removed to expose the skull. A craniotomy was performed on the left hemisphere along the midline and posterior sutures, covering an area of ~3.0×3.0 mm$^2$. The overlaying cortical tissues (including V1 and hippocampus) were removed by aspiration to expose the left SC. A head bar was finally mounted on the skull using Metabond (Parkell, Edgewood, NY) mixed with black ink. Animals previously injected with H2B-GCaMP6s would be ready for imaging. Animals to undergo imaging using the calcium-sensitive dye Cal520 would have the dye loaded into their SC as described below.

### Preparation and administration of the calcium-sensitive dye Cal-520

A fresh solution of the fluorogenic calcium-sensitive dye Cal-520 AM (AAT Bioquest; [*Tada et al., 2014*]) was prepared for every experiment. A solution of 20% Pluronic F-127 in DMSO was initially prepared and sonicated for 10–15 min. Four microliters of this solution were used to reconstitute 50 μg of powdered Cal-520. The resulting solution was sonicated for another 12–15 min and then brought to a total volume of 40 μl by adding 36 μl of a calcium-free solution (in mM: 150 NaCl, 2.5 KCl, and 10 HEPES, pH 7.4), for a final concentration of 1.13 mM Cal-520. After five more min of sonication, the solution was ready to be bolus loaded using a Nanoject II (Drummond) fitted with a glass pipette with a beveled tip and an inner diameter of 10–20 μm.

Once the SC was exposed, the pipette was filled with the previously prepared solution and lowered into the tissue. Twenty pulses of 2.3 nL each (46 nL total volume), at 20 s intervals, were administered to deliver the solution first at a depth of 450 μm below the surface, then at 200 μm after retracting the pipette to that depth. The pipette was left in the tissue for 1–2 min before being slowly retracted. The SC was then covered with ACSF (in mM: 125 NaCl, 5 KCl, 10 glucose, 10 HEPES, 2 CaCl$_2$, pH 7.4, 300 mOsm). Imaging was performed 1–2 hr after dye loading.

## Injection of H2B-GCaMP6s

Mice were anesthetized with isoflurane (5% for induction, 1.5% for maintenance, in $O_2$) then transferred onto a heating pad. Their body temperature was monitored via a rectal thermoprobe and maintained at 37°C through a feedback heater control module (Frederick Haer Company, Bowdoinham, Maine). Artificial tears (Henry Schein) were applied to the eyes for protection during surgery. The scalp was then shaved, and a small cut was made to expose the skull near the lambda point. A burr hole was drilled on the left hemisphere using a dental drill, 0.75 mm lateral and 0.5 mm anterior of the lambda point.

A Nanoject II (Drummond) fitted with a glass pipette with a beveled tip and an inner diameter of 10–20 µm, was used for viral injection. Viral particles were loaded into the pipette, which was then lowered into the brain through the burr hole, first to a depth of 1.4 mm below the pial surface, and then retracted to a depth of 1.2 mm. At each depth a total volume of roughly 50 nL was delivered, in 2.3 nL pulses, 15 s apart. AAV-syn-H2B-GCaMP6s (generously provided by Dr. Loren Looger, Janelia Research Campus) was injected into the SC of GAD2-Cre x AI9 (*RCL*-tdT) animals (1:1 in PBS). Alternatively AAV-syn-H2B-GCaMP6s was mixed with AAV1.CAG.Flex.tdTomato.WPRE.bGH ($5.1 \times 10^{12}$ GC/mL, University of Pennsylvania Vector Core, Allen Institute 864) (1:1:2 in PBS) and the same volumes were injected in the SC of GAD2-Cre mice at the aforementioned depths. The pipette was left in the tissue for 1–2 min before being slowly retracted. The skin was subsequently sutured back. Mice were given a dose of buprenex during surgery (0.05 mg/Kg, Sub-Q), and a dose of carprofen after (5 mg/Kg, Sub-Q), and were monitored daily for pain and wound health. Imaging was performed 10 to 21 days after injection.

## Two-photon calcium imaging

After the mice were prepared for imaging as described in the previous sections, they were moved onto a heating pad under a two-photon scanning microscope (2P-SGS or Ultima Investigator, Bruker Nano Surface Division). The head bar was clamped at an angle so that the imaged SC surface was largely flat and perpendicular to the optical axis of the objective. A thin film of silicone oil was applied to the eyes for protection. A shield was placed around the craniotomy to block light from the visual stimulus during imaging. The SC was covered with 3% agarose in ACSF for stability. Imaging was performed with a Ti:sapphire laser (Coherent Chameleon Ultra II) at excitation wavelengths of 800 nm for Cal-520, 920 nm for H2B-GCaMP6s, and 720 or 1020 nm for tdTomato, using a 40X, 0.8-NA Leica, or a 16X, 0.8-NA Nikon objective, immersed in ACSF. Emitted signals from the $Ca^{2+}$ indicators and tdTomato were filtered into separate PMTs (green and red channels). Laser excitation power after the objective was around 10 mW for Cal-520 imaging, and varied between roughly 10 and 120 mW (depending on the depth) for H2B-GCaMP6s imaging. With the 2P-SGS, data were acquired using PrairieView software (Versions 5.0 and 5.3) in spiral scan mode at 2X optical zoom, resulting in a circular field of view with a diameter of 135 µm. Image resolution was 256 × 256 pixels and the acquisition rate was 8.079 Hz. Data in *Figure 5* were acquired with the Ultima Investigator, using PrairieView software (Versions 5.4) with a resonant scanner at 4X optical zoom, resulting in a 206 × 206 µm field of view. Image resolution was 512 × 512 pixels and the acquisition rate was roughly 30 Hz. Frame-averaged data were used for the analysis (4-frame averages). Imaging with Cal-520 was performed in the superficial SGS (sSGS, no deeper than 50 µm from the SC surface), while imaging with H2B-GCaMP6s was performed across different depths of the SGS, ranging from the sSGS down to 205 µm below the surface.

## Visual stimulus for imaging

Visual stimuli were generated with Matlab Psychophysics toolbox (*Brainard, 1997*; *Niell and Stryker, 2008*) on an LCD monitor (37.5 cm ×30 cm, 60 Hz refresh rate,~50 cd/m² mean luminance, gamma corrected). The screen was placed 25 cm away from the eye contralateral to the imaging site (the right eye), and slightly tilted at an angle matching that of the mouse's head, given that the mouse's nose was slightly elevated to correct for the curvature of SC and allow imaging from a relatively flat surface. The monitor was moved for every imaged field of view so that the cells' receptive fields were near the center of the screen. The placement of the monitor center in visual space varied between 30° and −25° in elevation (0° representing eye-level) and between 30° and 90° across the

azimuth (0° representing the center of the binocular field) in all imaging experiments reported in this study. The ipsilateral eye was covered throughout the experiment.

Two types of visual stimuli were used for imaging. First, a flashing black square (5° x 5° in visual angle) on a gray background was used to map the receptive fields of the imaged neurons. The square was flashed in a 6 × 6 grid (30° x 30° in visual angle), for a duration of 1 s, followed by the presentation of a gray screen for 3 s. This stimulus set was shown to the mouse at least four times in a pseudo-random fashion for every imaged field of view.

The second visual stimulus was 'center-surround' square wave drifting gratings (100% contrast, 0.08 cpd, 2 Hz), presented on a gray background at the center of the screen so that the center component (20° across) covered the receptive fields of the imaged neurons. The surround was an annulus that started at the very edge of the center stimulus and extended 60° across. To assess responses to direction-contrast, eight different directions of motion were used for both center and surround gratings, ranging from 0° to 315° and tiling all of direction space in 45° increments. 0° represented forward motion from the animal's perspective; positive values followed in a clockwise fashion, and negative values in a counterclockwise fashion. A blank (gray) condition was added to the eight directions for both center and surround for a total of 81 (9 × 9) unique center-surround combinations (including center alone conditions, surround alone conditions, and a gray screen condition). To assess responses to phase contrast, the 8 directions of the center stimulus were coupled with an anti-phase surround (180° phase difference), moving in the same direction as the center, for a total of 8 unique conditions. To assess responses to temporal frequency contrast, the 8 directions of the center stimulus were coupled with a surround at an either lower (1 Hz), or higher (4 Hz) temporal frequency, moving in the same direction as the center, for a total of 16 unique conditions. Finally, to assess responses to orientation contrast, static gratings of 4 different orientations (100% contrast, 0.08 cpd) were used for both center and surround, ranging from 0° to 135° and tiling all of orientation space in 45° increments. 0° represented vertical orientation; positive values followed in a clockwise fashion, and negative values in a counterclockwise fashion. A blank (gray) condition was added to the four directions for both center and surround for a total of 25 (5 × 5) unique center-surround combinations (including center alone conditions, surround alone conditions, and a gray screen condition). Each stimulus condition of the gratings was presented for 2 s, followed by a gray screen for 5 s. Every stimulus set was shown to the mouse at least four times in a pseudo-random fashion for every imaged field of view.

## Imaging data analysis

Animals that had visible tissue damage to their SC after dye loading, where the dye failed to be incorporated into the cells, or where there was poor expression of H2B-GCaMP6s, were not subject to imaging. Data analysis was performed on all animals that were subject to imaging, and no data points were excluded from the resulting datasets.

Time-series frames were averaged to produce an average image of the field of view. In the cases where the imaging field shifted in the x-y plane over the course of the series, a semi-automated procedure was used to realign the frames. Specifically, a subset of the frames along the recording were manually realigned to match the first frame of the recording, and the corrected positions of all the intermediate frames were automatically extrapolated, leading to a sharper corrected average image, and spatially stable regions of interest (ROIs).

To determine whether each selected ROI is an inhibitory (GAD2$^+$) or excitatory neuron (GAD2$^-$), the experimenter referred to the red channel image of each field of view where GAD2$^+$ cells were labeled with tdTomato. This selection process relied exclusively on the expression of tdTomato and was performed blindly to the functional properties of the cells, which were determined at a later stage of the process.

For the analysis of Cal-520 imaging data, we followed our published procedures (*Inayat et al., 2015*). Briefly, ROIs were manually drawn on the average image of the collected time-series, and the intensity values of all pixels in each ROI were averaged for each frame to obtain the raw Ca$^{2+}$ signal for each cell. From the raw trace, and for each stimulus presentation, $\Delta F/F_0 = (F - F_0)/F_0$, was calculated, where $F_0$ was the mean of the baseline signal over a fixed interval (1.25 s for gratings; 0.75 s for flashing squares) before stimulus onset, and $F$ was the average fluorescence signal over a 2.5 s duration starting at 250 ms after stimulus onset and ending at 750 ms after stimulus offset for gratings (1.1 s duration, 250 ms after onset and 350 ms after offset, for flashing squares). A cell was

considered responsive if its mean $\Delta F/F_0$ was more than two standard derivations above its $F_0$ for at least one of the stimulus conditions. The mean value of $\Delta F/F_0$ for each of the stimulus conditions was then used to determine the direction tuning curve for every responsive cell, and to calculate a direction selectivity index and a response modulation index by the surround.

A similar procedure was used to analyze the H2B-GCaMP6s imaging data, with the exception that the 2.5 s integration time window of $\Delta F/F_0$ was shifted forward in time by 375 ms to account for the slower dynamics of H2B-GCaMP6s, compared to Cal-520. We analyzed this dataset with time windows of different latencies and durations, and our conclusions were not affected (*Figure 6—figure supplements 1* and *2*, and See 'Simultaneous two-photon imaging and cell-attached recording' below).

To quantify the degree of direction selectivity, we calculated a global direction selectivity index (gDSI), which is the vector sum of $\Delta F/F_0$ responses normalized by their scalar sum (*Gale and Murphy, 2014*; *Inayat et al., 2015*): $\mathrm{gDSI} = \frac{\sum R_\theta e^{i\theta}}{\sum R_\theta}$, where R$\theta$ is the response magnitude in $\Delta F/F_0$ at direction $\theta$ of the center stimulus.

To calculate the modulation index, we first determined each neuron's preferred direction as the center-stimulus direction that elicited the peak average $\Delta F/F_0$. In the case of neurons that were non-response to the center stimulus alone (center-silent neurons), the preferred direction was chosen as the center direction of the center-surround stimulus combination that elicited the peak average $\Delta F/F_0$. The modulation index was then calculated as follows:

$$\text{Modulation Index} = \frac{R_{\text{pref. C with S}} - R_{\text{pref. C}}}{R_{\text{pref. C with S}} + R_{\text{pref. C}}}$$

Where $R_{\text{pref. C with S}}$ is the neuron's response (in $\Delta F/F_0$) to coupling its preferred direction at the center with whichever surround we were assessing, and $R_{\text{pref. C}}$ is the neuron's response to the presentation of its preferred direction at the center alone. Negative numbers indicate a suppression by the surround of the response to the center alone, while positive numbers indicate potentiation. Values that were below $-1$ or above 1 due to negative $\Delta F/F_0$ values were adjusted to $-1$ and 1, respectively.

The RF center was determined by the following 'center of mass' equation, $RF\ Center,\ [x,y] = \frac{\sum R_i r_i}{\sum R_i}$, where $i$ represents the places in the grid where the cell was responsive. R and r represent the response magnitude ($\Delta F/F_0$) and position vector at the $i$th location, respectively (*Inayat et al., 2015*).

## Simultaneous two-photon imaging and cell-attached recording

We performed imaging-guided cell-attached recordings to characterize H2B-GCaMP6s and assess its capacity to report spiking activity. We used glass micropipettes (1.8–2.5 µm tip diameter, 2.2–6.5 M$\Omega$ tip resistance) filled with ACSF (in mM: 125 NaCl, 5 KCl, 10 glucose, 10 HEPES, and 2 CaCl$_2$, pH 7.4) and containing a mixture of 20 µM Alexa Fluor 488 and 594, for visualization under the microscope. Positive pressure was applied to the pipette, and the tip was brought to a position on top of a target neuron. The tip was subsequently lowered onto the cell, until a change in resistance was detected. Light suction was then applied to generate a seal and detect spiking activity. A Multi-Clamp 700B amplifier (Molecular Devices) in current-clamp mode and a System three workstation (Tucker-Davis Technologies) were used to record extracellular spiking. A minimal version of the Center-Surround stimulus (four directions and a blank in both center and surround) was used to elicit visual responses, and both image acquisition and the electrophysiological recording were synchronized to the visual stimulus.

The imaging and spiking data were then analyzed to compare their response magnitude to each stimulus condition (*Figure 6—figure supplement 1A–B*). Spike rate was averaged over the 2 s period of visual stimulus presentation (firing rate to the blank stimulus was subtracted). Due to its slow dynamics, the H2B-GCaMP6s calcium signal was averaged between 625 ms after stimulus onset and 1125 ms after offset for a duration of 2.5 s. This particular delay was longer than that used for Cal-520 (250 ms and 750 ms respectively). It was chosen so that for either reporter the start time coincided with a 20% response increase from baseline to the preferred C-S combination (*Figure 6—figure supplement 1C*, blue and red curves). Note that because of the slow dynamics of H2B-

GCaMP6s, and particularly the slow fluorescence decay time following a stimulus offset, a stable baseline was not always reached before the onset of the following stimulus. This resulted in negative $\Delta F/F_0$ values at some non-responsive conditions (*Figure 6—figure supplement 1B*). This could lead to an overestimation of the selectivity of cells, a foreseeable problem with calcium indicators that are slow or do not have single-spike resolution. Nevertheless, given the reasonably linear relationship between H2B-GCaMP6s and spike responses (n = 3, 2 mice; *Figure 6—figure supplement 1B, E*), relative comparisons of responses within a single tuning curve and between cells are still valid. Some neurons exhibited uncharacteristic activity when patched (n = 2, 2 mice), which resulted in the immediate saturation of their calcium signal; a phenomenon very rarely observed during regular imaging sessions, and clearly induced by the recording procedure. Those cells were excluded from further analysis (data not shown).

## Statistics

All pooled data were presented as mean ± s.e.m, unless stated otherwise. Significance was calculated using two-sided statistical tests including Mann-Whitney *U*-tests, Kolmogorov-Smirnov (K-S) tests, Wilcoxon signed-rank tests, and paired *t*-tests as stated. Correlation coefficients and their corresponding p-values were calculated, in addition to first degree polynomial fits and their corresponding $R^2$ values and y-intercepts, as mentioned in the text.

To determine a significant difference between a neuron's responses to any given pair of center-surround conditions (e.g. preferred-direction center vs. preferred center +oppositely moving surround), we performed a bootstrapping test. The four $\Delta F/F_0$ values (four trials) for each of the two compared stimulus conditions were pooled for a set of 8 values. Eight values were then sampled from that set 10,000 times, with replacement. Each of the generated sets was subsequently split into two subsets of 4 values randomly and the means of the two subsets subtracted to generate a distribution of the difference. The difference between the mean $\Delta F/F_0$ of the observed data was calculated, and depending on where that value fell with respect to the 95% confidence interval of the distribution, the response was considered significantly potentiated, suppressed, or unaltered. This measure gave us a more statistically tractable measure of modulation compared to the calculation of the modulation index, where a hard cutoff value of 0 separated between potentiated and suppressed neurons.

Note that when using this bootstrapping analysis we observe significant response suppression in center-silent neurons by a surround moving in the same direction, compared to their 'response' to the center alone (*Figure 2F*). This indicates that our method for assigning response significance is rather conservative, and that some of these neurons might actually have some low-magnitude response to the center stimulus alone.

All analyses and graph plotting were performed in Matlab (MathWorks). The Matlab workspace used to generate *Figures 1–7* are included in the three source data files. No statistical methods were used to predetermine sample sizes, but our sample sizes are similar to those reported in the field. We did not randomly assign animals to groups because it is not applicable to the experimental design of this study.

## Ackowledgements

The authors wish to thank Dr. Loren Looger and Dr. Na Ji from the Janelia Research Campus for providing H2B-GCaMP6s. This work was supported by US National Institutes of Health (NIH) grants (EY026286 and EY020950) to JC. National Natural Science Foundation of China (NSFC) grant (81770956) and Tianjin Science Fund for Distinguished Young Scholars (17JCJQJC46000) to XS.

## Additional information

### Funding

| Funder | Grant reference number | Author |
| --- | --- | --- |
| National Institutes of Health | R01EY020950 | Jianhua Cang |
| National Institutes of Health | R01EY026286 | Jianhua Cang |

| National Natural Science Foundation of China | 81770956 | Xuefeng Shi |
| TIanjin Science Fund for Distinguished Young Scholars | 17JCJQJC46000 | Xuefeng Shi |

The funders had no role in study design, data collection and interpretation, or the decision to submit the work for publication.

## Author contributions

Jad Barchini, Conceptualization, Performed all imaging experiments and analyzed the data, Made all figures, Wrote the paper; Xuefeng Shi, Writing—review and editing, Performed the cell-attached recordings and analyzed the data; Hui Chen, Software, Writing—review and editing; Jianhua Cang, Conceptualization, Supervision, Funding acquisition, Project administration, Writing—review and editing, Helped with the imaging experiments and analysis

## Author ORCIDs

Jianhua Cang (iD) https://orcid.org/0000-0002-0760-7468

## Ethics

Animal experimentation: All experimental procedures were approved by the Northwestern University Institutional Animal Care and Use Committee, protocol #IS00001946.

## Decision letter and Author response

Decision letter https://doi.org/10.7554/eLife.35261.021
Author response https://doi.org/10.7554/eLife.35261.022

# Additional files

## Supplementary files

• Source data 1. The Matlab workspace used to generate *Figures 1–4* are included in this ZIP file.
DOI: https://doi.org/10.7554/eLife.35261.016

• Source data 2. The Matlab workspace used to generate *Figure 5* are included in this ZIP file.
DOI: https://doi.org/10.7554/eLife.35261.017

• Source data 3. The Matlab workspace used to generate *Figures 6–7* are included in this ZIP file.
DOI: https://doi.org/10.7554/eLife.35261.018

• Transparent reporting form
DOI: https://doi.org/10.7554/eLife.35261.019

## Data availability

Source data files have been provided for Figures 1–7.

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
