## [Decision Letter]

Thank you for submitting your article "Bidirectional Encoding of Motion Contrast in the Mouse Superior Colliculus" for consideration by *eLife*. Your article has been reviewed by 3 peer reviewers, Fred Rieke as the Reviewing Editor, and Andrew King as the Senior Editor. The reviewers have opted to remain anonymous.

The reviewers have discussed the reviews with one another and the Reviewing Editor has drafted this decision to help you prepare a revised submission.

The reviewers were enthusiastic and felt that the work potentially makes an important and timely contribution. At the same time, several issues need to be strengthened before we can reach a final decision. These include:

1) It is unclear how much of the results can be accounted for by the classic receptive field, and which require responses outside the classic receptive field.

2) The interpretation of the results needs to be discussed more carefully, with an emphasis on what is established and what is speculative. This should include clarifying that the origin of the measured signals and the connection to saliency are both speculative.

3) Additional analyses are needed to establish definitively that suppression is directionally tuned in excitatory but not inhibitory neurons.

These and other issues are detailed in the individual reviews below.

Reviewer #1:

This paper describes the encoding of motion in mouse SC, with a particular focus on how motion in the receptive field center and surround interact. The work is technically impressive and the results quite clear. My concerns are largely to do with the presentation and the strength of the conceptual advance of the work described.

Relation of results to saliency.

Throughout the paper, the results are described as a saliency computation. But unless I am missing something, the effects are close to fairly standard receptive field descriptions. Thus, for a cell sensitive to luminance, responses are suppressed with center and surround are stimulated with the same polarity inputs, and responses are potentiated with opposite polarity inputs. The current paper describes a similar receptive field behavior for motion. I would find the paper easier to read if this was described in terms of local vs. global motion rather than in terms of saliency. One issue I found particularly hard to relate the results of the paper to, was the idea of a saliency map.

Reviewer #2:

The manuscript provides a very readable, well-illustrated and interesting report on data from the superficial (i.e. visual sensory) layers of the superior colliculus in anaesthetised mouse. The authors use 2p imaging to measure calcium responses in inhibitory (GAD2 expressing) and other neurons. They report responses to stimuli composed of a drifting square-wave central grating, and a surround grating that could drift at the same rate in the same direction, or other directions, different drift rates, or different phase.

The experiments appear to be very well performed, and the analyses are clearly careful and well thought through. The current manuscript contributes significantly to our understanding of SC function by distinguishing excitatory from inhibitory neurons, and by showing that there is a depth dependence to these effects. The data may also be important for understanding the 'building blocks' of saliency computations.

The claims are as follows:

1) In very superficial excitatory neurons surrounds of the same motion direction suppress, but those of the opposite direction strongly facilitate. Strong facilitation is not seen for surrounds of different phase or temporal frequency.

2) Inhibitory neurons throughout layers instead show 'pan-direction' suppression.

3) There is reduced opposite-direction facilitation in deeper excitatory neurons.

Claim 1: The measurements in excitatory neurons show the expected suppression by surrounds moving in the same direction alongside an unexpected and powerful facilitation by oppositely moving surrounds. This is a very intriguing result and I would like to see some more evidence that this is not interactions among stimuli within the classical receptive field (if the surround is strongly suppressive, it could mask responses from the parts of the receptive field that it lies over, so lack of response to the surround alone needs to be treated with caution). I make this statement because the experiments in Figure 4 show that the suppression is reduced by manipulating surround drift rate or phase (i.e. the effect of these manipulations is in the same direction as the opposite-direction facilitation). Drift in opposite direction may have stronger impact than offset spatial phase and some analysis of models of direction-selectivity would be useful to be secure that the pattern of effects could not be attributed to interactions with DS receptive fields.

Claim 2: The measurements of surround suppression in inhibitory neurons are harder to interpret, as the authors point out, because the 'surround alone' stimulus drove responses strongly in many or most neurons. This probably would not change the overall conclusion that inhibitory neurons show suppression but not facilitation, but it certainly may impact on the estimates of direction sensitivity of surround suppression. More analyses are needed to be more secure in this. For example, the authors compare response to center stimulus to response to center + same surround stimulus (Figure 3B) but I think it more appropriate (here and in other analyses) to compare (center alone + surround alone) to the center + surround stimulus.

Claim 3: That deeper layers show less opposite direction facilitation is very intriguing. The analyses here however aren't quite as comprehensive as those above – for example I cannot see results for same direction surround, and therefore find it difficult to know whether the depth dependence reflects specific differences in direction-dependent suppression or also non-direction dependent suppression. It would also be useful to consider the impact of changes in receptive field size at deeper depths (Ahmadlou et al., 2017).

Reviewer #3:

Contextual features in visual scenes have large effects on sensory processing and behavior. In particular, these features have been suggested to work through fast, bottom-up mechanisms to alter the saliency of a stimulus. However, the mechanisms that support these effects, and the areas that are responsible, are not well understood. In this manuscript, Barchini et al., address this question by imaging the activity of identified GAD2^+^ and GAD2^-^ neurons in the mouse superior colliculus (SC), an area thought to be involved in the bottom-up contextual processing. They present a rich dataset in which they cleverly explore stimulus space and find that GAD2^-^ neurons in the superficial layers of the SC undergo bidirectional contextual modulation. Altogether, these data are clear, beautifully quantified and reveal novel relationships between context and sensory processing in the SC. My only major request is that the authors clarify their proposed SC-intrinsic model for bidirectional contextual modulation.

Essential revisions:

The authors argue (in the Abstract, Results section and Discussion section) that the bidirectional contextual modulation of GAD2^-^ activity is driven by local interactions within the SC. However, there remain a number of confusing points about this model:

a) First, it is not clear what is driving the suppression of the GAD2^+^ cells when the center and surround move in the opposite direction. In order for this to be a SC-intrinsic mechanism, there must be a population of GAD2^+^ cells that is potentiated by opposite direction stimuli (to suppress the high DSI GAD2^+^ cells), but these are rarely observed. Is it proposed that the potentiated neurons in Figure 3—figure supplement 1F are the source of this inhibitory drive to the GAD2^+^ cells? Alternatively, is the proposal that this is mediated by longer-range inputs from GAD2^+^ cells centered on the surround of the studied neurons? The authors need to clarify their model for how high DSI GAD2^+^ cells are selectively suppressed by opposite center-surround stimuli.

b) Second, it is not clear what is driving the suppression of all cells when the center and surround move in the same direction. Again, if this is a SC-intrinsic mechanism, there must be a population of GAD2^+^ cells that is potentiated by these stimuli. However, these are very rarely observed. Again, is this proposed to be a long-range mechanism? Or is it possible that the potentiation and suppression are supported by distinct mechanisms where the latter is inherited from the retina? The authors need to clarify their model for how same-direction stimuli suppress most SC cells.

c) Third, it is not clear what drives the depth-dependence of the observed effects. Despite the change in DSI with depth, the GAD2^+^ cells are similarly suppressed by opposite direction stimuli at all depths. Thus, the suppression of these cells should be sufficient to drive potentiation of GAD2^-^ cells at all depths. Is there a specific model of connectivity that the authors suggest can account for this?

d) Fourth, it is not clear why this effect is specific to direction. The authors demonstrate that cross-orientation stimuli are generally suppressive (though less suppressive than iso-orientation stimuli) in both GAD2^-^ and GAD2^+^ cells. They also show that when the center and surround are moving in the same direction, but antiphase or at different TFs, this disrupts the suppressive nature of the surround, though does not drive potentiation. How do these data fit into the authors' model for contextual modulation?

e) Finally, something strange is happening in Figure 3—figure supplement 1B. There seems to be a facilitation of GAD2^+^ neurons' responses to the non-preferred direction when the preferred direction is in the surround, making these neurons orientation tuned. Is this true for individual neurons (are they orientation tuned in this context) or do some neurons start responding selectively to the opposite direction? Moreover, how does this result fit into the authors’ model of contextual modulation in the SC?

[Editors' note: further revisions were requested prior to acceptance, as described below.]

Thank you for resubmitting your work entitled "Bidirectional Encoding of Motion Contrast in the Mouse Superior Colliculus" for further consideration at *eLife*. Your revised article has been favorably evaluated by Andrew King (Senior Editor) and a Reviewing Editor.

This is a revision of a paper on motion coding in superior colliculus. The authors have done a good job responding to comments from the previous round of reviews, but there are some points that need to be addressed before acceptance. A few remaining issues and a few new ones that were clarified from the revised paper follow.

Reviewing editor comments:

OMS ganglion cells

The encoding of local motion described here resembles in several ways that of object-motion sensitive ganglion cells – see Olveczy et al., (2003), Baccus et al., (2008). The similarities and any dissimilarities should be described.

Subsection “Responses of sSGS excitatory neurons are modulated by motion contrast”: What criterion was used to determine which cells were sufficiently well centered to include in the analysis? This is related to the request in the previous round of reviews to show individual receptive fields. I think those included in the response to reviews should be included as a figure supplement.

Subsection “Responses of sSGS excitatory neurons are modulated by motion contrast”: I think this is the first place bidirectional modulation comes up. Please define that term here.

Subsection “Inhibitory neurons in the sSGS are suppressed by motion contrast” and elsewhere: statistical quantities often have too many digits – e.g. here the p value could be 6.0e^-20^.

Subsection “Inhibitory neurons in the sSGS are suppressed by motion contrast”: Is the surround for inhibitory neurons clearly a surround (e.g. do you see the surround with flashed squares, or could the RFs of excitatory and inhibitory neurons be quite different in size)?

Subsection “Inhibitory neurons in the sSGS are suppressed by motion contrast”: Sentence is awkward and could get rewritten or broken up.

p-values: Make sure it is clear for each use what is being tested.

Subsection “Inhibitory neurons in the sSGS are suppressed by motion contrast”: Please spell this suggestion out in a bit more detail.

Subsection “sSGS are specifically tuned to motion direction contrast”: The second paragraph is hard to follow, particularly relationship between text and figure.

Subsection “sSGS are specifically tuned to motion direction contrast”: Red dashed line?

Subsection “Depth-dependent motion contrast coding in the SGS”: Why is the last paragraph in this section about depth dependence? It feels out of place.

Discussion section: The jump from orientation to motion in this paragraph is a bit disconcerting. One possibility is to return to the pop out phenomena mentioned in the Introduction.

Subsection “Mechanisms for motion contrast computation in the mouse SGS”: This section could get tightened considerably.

---

## [Author Response]

The reviewers were enthusiastic and felt that the work potentially makes an important and timely contribution. At the same time, several issues need to be strengthened before we can reach a final decision. These include:1) It is unclear how much of the results can be accounted for by the classic receptive field, and which require responses outside the classic receptive field.

We agree that much of our results are consistent with the known properties of classical receptive fields from studies in visual cortex in primates and cats. What’s new of our finding is that (1) We were the first to study motion contrast coding in the mouse SC, a new model in vision research; (2) We found bidirectional and monotonic encoding of motion contrast in this structure; and (3) Most importantly, we found striking difference between the responses of excitatory and inhibitory neurons to motion contrast, the like of which had not been revealed in any structure of any species. We have made a number of changes in Abstract, Introduction, and Discussion section to address this issue.

2) The interpretation of the results needs to be discussed more carefully, with an emphasis on what is established and what is speculative. This should include clarifying that the origin of the measured signals and the connection to saliency are both speculative.

We have followed these comments and expanded discussion. Specifically, we have (1) Strengthened the argument why the observed bidirectional encoding of motion contract requires local collicular circuits; (2) Speculated what circuit component might be at play; and (3) Clarified how our finding is related to saliency computation.

3) Additional analyses are needed to establish definitively that suppression is directionally tuned in excitatory but not inhibitory neurons.As detailed below, we have performed a number of new analyses following the reviewers’ suggestions. Many new panels are added to the main and supplemental figures, and several are included in this response letter. All these new analyses support our original conclusions.Reviewer #1:This paper describes the encoding of motion in mouse SC, with a particular focus on how motion in the receptive field center and surround interact. The work is technically impressive and the results quite clear. My concerns are largely to do with the presentation and the strength of the conceptual advance of the work described.Relation of results to saliency.Throughout the paper, the results are described as a saliency computation. But unless I am missing something, the effects are close to fairly standard receptive field descriptions. Thus, for a cell sensitive to luminance, responses are suppressed with center and surround are stimulated with the same polarity inputs, and responses are potentiated with opposite polarity inputs. The current paper describes a similar receptive field behavior for motion. I would find the paper easier to read if this was described in terms of local vs global motion rather than in terms of saliency. One issue I found particularly hard to relate the results of the paper to, was the idea of a saliency map.

We agree with the reviewer that our findings are consistent with the known properties of classical receptive fields in terms center-surround interactions. As we stated in Introduction, the standard description of RF center-surround interaction is an important part of saliency computation. Consistently, our study is indeed about local vs. global motion, i.e., different motion contrasts between center and surround stimuli. In other words, it is about how mouse SC neurons encode motion stimuli of different saliency values: more salient when center and surround moving in different directions.

Therefore, our study does not directly address the saliency “map”, but instead, it reveals a 'building block' of saliency computation (reviewer #2). More specially, it is about the encoding of visual motion saliency. We have made a number of textual changes in Abstract, Introduction and Discussion section to clarify this issue.

Reviewer #2:[…] Claim 1: The measurements in excitatory neurons show the expected suppression by surrounds moving in the same direction alongside an unexpected and powerful facilitation by oppositely moving surrounds. This is a very intriguing result and I would like to see some more evidence that this is not interactions among stimuli within the classical receptive field (if the surround is strongly suppressive, it could mask responses from the parts of the receptive field that it lies over, so lack of response to the surround alone needs to be treated with caution). I make this statement because the experiments in Figure 4 show that the suppression is reduced by manipulating surround drift rate or phase (i.e. the effect of these manipulations is in the same direction as the opposite-direction facilitation). Drift in opposite direction may have stronger impact than offset spatial phase and some analysis of models of direction-selectivity would be useful to be secure that the pattern of effects could not be attributed to interactions with DS receptive fields.

This is an interesting possibility, but we think it’s unlikely for several reasons. First, if the “surround” stimulus indeed covers some of the classical RF, the cell could respond even more when the surround of “same” direction is coupled with the center stimulus at the cells’ preferred direction (i.e., the surround is also at the cell’s preferred direction), compared to “preferred center + opposite surround” (i.e., when the surround is at the cell’s null direction). But the opposite was actually observed: preferred center + opposite surround induced powerful facilitation. This indicates that, even if the surround covered some of the classical RF, it should not contribute to the observed facilitation. Second, the dramatic difference between excitatory and inhibitory cells in response to “center + opposite surround” argues against the possibility of “interactions among stimuli”, as that would cause similar changes in both cell types. Finally, our collaborator Dr. Wei Wei at University of Chicago used similar center-surround stimuli as ours to probe retinal responses. Their results strongly demonstrate that the facilitation we see in the SC excitatory cells are generated by local collicular circuits, thus not due to stimulus interactions.

Claim 2: The measurements of surround suppression in inhibitory neurons are harder to interpret, as the authors point out, because the 'surround alone' stimulus drove responses strongly in many or most neurons. This probably would not change the overall conclusion that inhibitory neurons show suppression but not facilitation, but it certainly may impact on the estimates of direction sensitivity of surround suppression. More analyses are needed to be more secure in this. For example, the authors compare response to center stimulus to response to center + same surround stimulus (Figure 3B) but I think it more appropriate (here and in other analyses) to compare (center alone + surround alone) to the center + surround stimulus.

We have performed the suggested analysis and generated new plots to compare “center alone + surround alone” to the response to “center + surround stimulus”. We did this for both excitatory (not shown) and inhibitory cells (Figure3 —figure supplement 1C-D). The only major difference is the even stronger suppression by the same surround stimulus in inhibitory neuron, which is expected due to the addition of the preferred-surround-only responses to the x-axis.

Claim 3: That deeper layers show less opposite direction facilitation is very intriguing. The analyses here however aren't quite as comprehensive as those above – for example I cannot see results for same direction surround, and therefore find it difficult to know whether the depth dependence reflects specific differences in direction-dependent suppression or also non-direction dependent suppression.

As suggested, we have added these panels to Figure 6 (panel D and G). Due to the added panels, we have split this Figure (now Figures 6 and Figure 7).

It would also be useful to consider the impact of changes in receptive field size at deeper depths (Ahmadlou et al., 2017).

Our imaging is limited to the top 200 μm in depth, where there doesn’t seem to be much variability in RF size, consistent with a recent paper (Ito et al., 2017). RF sizes start to become noticeably larger below 200 um, a depth which we could not sample with calcium imaging. We therefore are not in a position to perform this analysis.

Reviewer #3:
*Contextual features in visual scenes have large effects on sensory processing and behavior. In particular, these features have been suggested to work through fast, bottom-up mechanisms to alter the saliency of a stimulus. However, the mechanisms that support these effects, and the areas that are responsible, are not well understood. In this manuscript, Barchini et al., address this question by imaging the activity of identified GAD2*^+^
*and GAD2*^-^
*neurons in the mouse superior colliculus (SC), an area thought to be involved in the bottom-up contextual processing. They present a rich dataset in which they cleverly explore stimulus space and find that GAD2*^-^ neurons in the superficial layers of the SC undergo bidirectional contextual modulation. Altogether, these data are clear, beautifully quantified and reveal novel relationships between context and sensory processing in the SC. My only major request is that the authors clarify their proposed SC-intrinsic model for bidirectional contextual modulation.Essential revisions:

*The authors argue (in the Abstract, Results section and Discussion section) that the bidirectional contextual modulation of GAD2*^-^
*activity is driven by local interactions within the SC. However, there remain a number of confusing points about this model:*

This concern as illustrated below in the 5 detailed points is all about the mechanisms of the observed bidirectional contextual modulation of GAD2^-^ activity. We have tried to address the following 5 issues the best we can, but we simply do not know the exact circuit diagram to explain all aspects of this finding. However, we are very convinced that much of the observed modulation is due to local interactions in the SC. This is based on findings from our collaborator Dr. Wei Wei at University of Chicago. They used very similar center-surround stimuli as in our experiments so that we can compare surround modulation from RGCs to the SC.

The main finding in the retina is that the direction selective retinal ganglion cells (DSGCs) are greatly suppressed by same direction surround, and the suppression is reduced (but no potentiation) when the surround is antiphase or moving along opposite direction. The fact that opposite and antiphase surround cause similar responses in DSGCs, but very different responses in SC, strongly argue for a role of local circuits in this computation. We have mentioned some of the retinal findings in our Discussion section to make this point clearer.

*a) First, it is not clear what is driving the suppression of the GAD2*^+^
*cells when the center and surround move in the opposite direction. In order for this to be a SC-intrinsic mechanism, there must be a population of GAD2*^+^
*cells that is potentiated by opposite direction stimuli (to suppress the high DSI GAD2*^+^
*cells), but these are rarely observed. Is it proposed that the potentiated neurons in Figure 3—figure supplement 1F are the source of this inhibitory drive to the GAD2*^+^
*cells? Alternatively, is the proposal that this is mediated by longer-range inputs from GAD2*^+^
*cells centered on the surround of the studied neurons? The authors need to clarify their model for how high DSI GAD2*^+^
*cells are selectively suppressed by opposite center-surround stimuli.*

The reviewer is correct that there has to be an increased inhibitory drive to mediate the suppression of GAD2^+^ cells when the center and surround move in opposite directions. We think it’s likely that this is mediated by long-range inhibition from the surround onto local inhibitory neurons. In other words, it’s from inhibitory neurons that are outside the imaged region. This is consistent with a recent finding by Kasai and Isa, 2016, which support the long range inhibition model. We have mentioned this in Discussion section.

*b) Second, it is not clear what is driving the suppression of all cells when the center and surround move in the same direction. Again, if this is a SC-intrinsic mechanism, there must be a population of GAD2*^+^
*cells that is potentiated by these stimuli. However, these are very rarely observed. Again, is this proposed to be a long-range mechanism? Or is it possible that the potentiation and suppression are supported by distinct mechanisms where the latter is inherited from the retina? The authors need to clarify their model for how same-direction stimuli suppress most SC cells.*

As mentioned above, the global suppression by the same direction stimulus is most likely inherited from the retina. However, precise local circuits must exist to generate the potentiation in excitatory cells when the center and surround stimuli move in opposite directions. In other words, it is possible for these two phenomena to arise through distinct mechanisms: one feed-forward from the retina, and the other through local activity in the SC. We have clarified this in the revised Discussion section.

*c) Third, it is not clear what drives the depth-dependence of the observed effects. Despite the change in DSI with depth, the GAD2*^+^
*cells are similarly suppressed by opposite direction stimuli at all depths. Thus, the suppression of these cells should be sufficient to drive potentiation of GAD2*^-^
*cells at all depths. Is there a specific model of connectivity that the authors suggest can account for this?*

This is indeed an intriguing finding. We do not have a specific model to explain the depth dependence effect, but we think it is informative to report this finding. However, it is relevant to note that just the suppression of inhibitory cells would not necessarily lead to potentiation in excitatory cells, because the excitatory retinal drive is also reduced at the same time. This consideration further supports a role of local circuits in this computation, and also suggests that such circuits would be lamina-specific. How such circuits are organized in different depths is completely unknown, and at this stage we can only speculate given the available data. We have mentioned this issue in Discussion section.

*d) Fourth, it is not clear why this effect is specific to direction. The authors demonstrate that cross-orientation stimuli are generally suppressive (though less suppressive than iso-orientation stimuli) in both GAD2*^-^
*and GAD2*^+^
*cells. They also show that when the center and surround are moving in the same direction, but antiphase or at different TFs, this disrupts the suppressive nature of the surround, though does not drive potentiation. How do these data fit into the authors' model for contextual modulation?*

This is indeed the heart of this question. As mentioned above, some of our findings can be explained by changes in the retina, but not for the direction-specific potentiation. A possible circuit is that long-range excitation and inhibition from the surround could be organized in the direction-specific manner. We are actively pursuing studies to reveal underlying circuits, but all we can do now in this paper is to speculate possible mechanisms. We have expanded this section of Discussion section.

*e) Finally, something strange is happening in Figure 3—figure supplement 1B. There seems to be a facilitation of GAD2*^+^
*neurons' responses to the non-preferred direction when the preferred direction is in the surround, making these neurons orientation tuned. Is this true for individual neurons (are they orientation tuned in this context) or do some neurons start responding selectively to the opposite direction? Moreover, how does this result fit into the authors’ model of contextual modulation in the SC?*

The apparent orientation tuning is an “artifact” of our choice of the center-stimulus size. The center stimulus size captures very well the response of excitatory neurons, which are rarely and weakly driven by any stimulus in the surround. Many inhibitory neurons, on the other hand, are significantly driven by the surround grating alone (for which we speculated possible causes in the text). Thus, when an inhibitory cell is presented by its null direction at the center coupled with the opposite surround, the surround direction is actually at the cell’s preferred direction (+180 in the red tuning curve) and would induce significant responses. In other words, the peak of the blue tuning curve in Figure 3—figure supplement 1B accounts for the response observed at the tail end of the red tuning curve. As explained above, long-range inhibition and excitation are more likely to be involved in the observed effect, this result in question is unlikely a factor.

[Editors' note: further revisions were requested prior to acceptance, as described below.]Reviewing editor comments:OMS ganglion cellsThe encoding of local motion described here resembles in several ways that of object-motion sensitive ganglion cells – see Olveczy et al., (2003), Baccus et al., (2008). The similarities and any dissimilarities should get described.

We have now cited and discussed these two studies (Subsection “Mechanisms for motion contrast computation in the mouse SG”).

Subsection “Responses of sSGS excitatory neurons are modulated by motion contrast”: What criterion was used to determine which cells were sufficiently well centered to include in the analysis? This is related to the request in the previous round of reviews to show individual receptive fields. I think those included in the response to reviews should be included as a figure supplement.

We have added one figure supplement to show these example RFs (Figure 1—figure supplement 1). The criterion has more to do with the center-surround stimulus than RF mapping. Cells were included if they were “center-responsive”, because some cells were responsive without having mappable RFs and we did not want to exclude those cells. RF maps for individual fields of view were visually assessed to make sure that we were not grossly off-centered in relation to the “center” drifting gratings stimulus. For “center-silent” cells, we made sure to exclude those that were responsive to the presentation of the surround-alone stimulus, as this would indicate an off-centered effective RF in relation to our visual stimulus. This was mentioned in the text (subsection “Responses of sSGS excitatory neurons are modulated by motion contrast”). In the paper, we reported the number of cells that were activated by the surround alone (few excitatory cells and more inhibitory cells), and performed additional analysis only including inhibitory cells that did not show any significant response to the surround alone stimulus and examined their response profiles (subsection “Inhibitory neurons in the sSGS are suppressed by motion contrast”, Figure 3 – —figure supplement 1E).

Subsection “Responses of sSGS excitatory neurons are modulated by motion contrast”: I think this is the first place bidirectional modulation comes up. Please define that term here.

Done.

Subsection “Inhibitory neurons in the sSGS are suppressed by motion contrast” and elsewhere: statistical quantities often have too many digits – e.g. here the p value could be 6.0e-20.

We have made these changes.

Subsection “Inhibitory neurons in the sSGS are suppressed by motion contrast”: Is the surround for inhibitory neurons clearly a surround (e.g. do you see the surround with flashed squares, or could the RFs of excitatory and inhibitory neurons be quite different in size)?

This issue was addressed in the second paragraph of the following page. No difference between RF size was seen between excitatory and inhibitory neurons (subsection “Inhibitory neurons in the sSGS are suppressed by motion contrast”).

Subsection “Inhibitory neurons in the sSGS are suppressed by motion contrast”: Sentence is awkward and could get rewritten or broken up.

We have followed the suggestion and broken it up into 2 sentences.

p-values: Make sure it is clear for each use what is being tested.

We have added such information when needed.

Subsection “Inhibitory neurons in the sSGS are suppressed by motion contrast”: Please spell this suggestion out in a bit more detail.

We have changed this sentence to make it clearer.

Subsection “sSGS are specifically tuned to motion direction contrast”: The second paragraph is hard to follow, particularly relationship between text and figure.

We have improved the writing in this paragraph to make it easier to follow.

Subsection “sSGS are specifically tuned to motion direction contrast”: Red dashed line?

Corrected.

Subsection “Depth-dependent motion contrast coding in the SGS”: Why is the last paragraph in this section about depth dependence? It feels out of place.

We have added a sentence to improve the transition.

Discussion section: The jump from orientation to motion in this paragraph is a bit disconcerting. One possibility is to return to the pop out phenomena mentioned in the Introduction.

We have followed this suggestion and improved the writing.

Subsection “Mechanisms for motion contrast computation in the mouse SGS”: This section could get tightened considerably.

We have tightened this section by removing several points that are not quite as necessary.